# Erase or Hide? Suppressing Spurious Unlearning Neurons for Robust Unlearning

**Nakyeong Yang**[1,2*]      **Dong-Kyum Kim**[2]      **Jea Kwon**[2]
**Minsung Kim**[1]      **Kyomin Jung**[1]      **Meeyoung Cha**[2†]
[1]Seoul National University      [2]Max Planck Institute for Security and Privacy
{yny0506, kms0805, kjung}@snu.ac.kr
{dong-kyum.kim, jea.kwon, mia.cha}@mpi-sp.org

## Abstract

Large language models trained on web-scale data can memorize private or sensitive knowledge, raising significant privacy risks. Although some unlearning methods mitigate these risks, they remain vulnerable to "relearning" during subsequent training, allowing a substantial portion of forgotten knowledge to resurface. In this paper, we show that widely used unlearning methods cause *shallow alignment*: instead of faithfully erasing target knowledge, they generate *spurious unlearning neurons* that amplify negative influence to hide it. To overcome this limitation, we introduce Ssiuu, a new class of unlearning methods that employs attribution-guided regularization to prevent spurious negative influence and faithfully remove target knowledge. Experimental results confirm that our method reliably erases target knowledge and outperforms strong baselines across two practical retraining scenarios: (1) adversarial injection of private data, and (2) benign attack using an instruction-following benchmark. Our findings highlight the necessity of robust and faithful unlearning methods for safe deployment of language models.

## 1 Introduction

Large language models (LLMs) are built on vast corpora of web-scale data, equipping them with broad capabilities across diverse tasks. Yet, this scale introduces privacy risks, as training datasets may inadvertently contain sensitive or personally identifiable information. In response, prior works have explored strategies to remove private or sensitive knowledge from LLMs. Such approaches include gradient-based interventions (Jang et al., 2022; Maini et al., 2024), preference-driven optimization frameworks (Jin et al., 2024; Yang et al., 2025), and representation learning techniques (Li et al., 2024), each of which aims to mitigate privacy risks embedded in model parameters.

Despite these efforts, prior studies reveal that existing unlearning techniques often fail to robustly eliminate target knowledge. Models subjected to such interventions remain susceptible to prompt-based elicitation (Jin et al., 2024; Yang et al., 2025) and can inadvertently recover forgotten information through representational shifts introduced by subsequent training (Deeb & Roger, 2024; Hu et al., 2024). Given the growing prevalence of open-source LLMs (e.g., Meta's Llama (Dubey et al., 2024), Alibaba's Qwen (Yang et al., 2024a)) and the widespread availability of fine-tuning interfaces, it is crucial to understand their vulnerabilities and design resilient unlearning methods fit for real-world deployment.

This paper shows that current unlearning methods tend to induce *shallow alignment*, a phenomenon where target knowledge is obscured but not truly removed. To explain this behavior, we propose the novel concept of *spurious unlearning neurons* and demonstrate that existing methods often suppress the target by introducing distinct neurons to act as inhibitors—i.e., spurious unlearning neurons—rather than diminishing the influence of neurons that truly encode sensitive information. Since the original knowledge-bearing neurons remain intact, the target information can re-emerge if these spurious neurons are disrupted or bypassed during subsequent training, ultimately leading to unlearning failures. Therefore, we argue that, for robust unlearning, methods should directly erase the true knowledge representations while preventing the emergence of spurious unlearning neurons.

---

* This work was done at the Max Planck Institute for Security and Privacy
† Corresponding author: mia.cha@mpi-sp.org

To illustrate this issue, we first investigate whether widely used unlearning methods can effectively remove target knowledge using an explainability method. Specifically, we apply an attribution method (Yang et al., 2023) to examine variations in neuronal contributions to target knowledge and compare variations in positive and negative influences before and after unlearning. If the knowledge is effectively removed, the positive influence on the target knowledge should diminish after the unlearning process. However, our experiments consistently show that the positive influence is retained while the negative influence increases. Therefore, these results suggest that widely used unlearning methods do not reliably eliminate the underlying knowledge encoded in the parameters; instead, they introduce new neurons to suppress it.

To better understand the shallow alignment problem, we introduce two practical attack scenarios to evaluate whether unlearned knowledge is truly removed or re-emerges during subsequent training: (1) adversarial injection via fine-tuning with a privacy-related dataset and (2) benign attack using an instruction-following benchmark. In the former case, an unlearned model is retrained with a small set of private data samples. If additional unlearned knowledge resurfaces from retraining on a disjoint private dataset, this implies that target knowledge has not been adequately removed. In the latter case, an unlearned model is retrained with a benign dataset, such as instruction-following data (e.g., Alpaca). If unlearned knowledge is recovered during this process, it poses significant privacy risks. Our experiments show that existing widely used unlearning methods are vulnerable to both attacks; as a result, the unlearning effect is compromised and target knowledge is easily recovered.

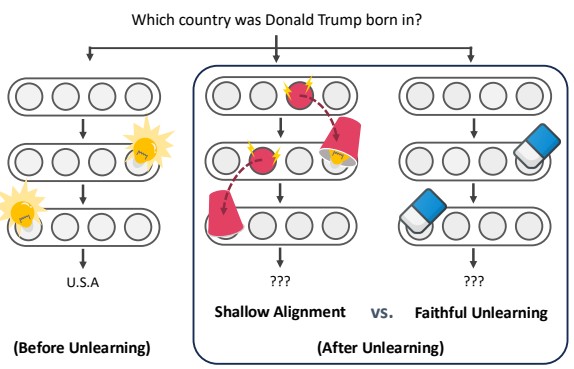

Figure 1: **Shallow Unlearning Alignment.** The red-colored neurons are *spurious unlearning neurons*, which hide target knowledge rather than erasing it. This presents a critical issue, since the knowledge may resurface if this shallow alignment is weakened or bypassed.

To address persistent limitations in existing unlearning methods, we propose SSIUU—**S**uppressing **S**pur**i**ous **U**nlearning Neurons for Robust **U**nlearning—which regularizes the increase of negative influence and thus enables unlearning algorithms to erase target knowledge effectively rather than suppress it. Specifically, we compute the attribution score for target knowledge and constrain the negative attribution values to remain at their original levels. Experimental results show that SSIUU outperforms strong baselines in the two practical attack scenarios, faithfully removing target knowledge and remaining robust to further retraining. We further analyze internal attribution signals, showing that SSIUU achieves robust unlearning by preventing the emergence of spurious unlearning neurons. Our method faithfully decreases the positive influence on target knowledge across all layers, while suppressing the growth of negative influence. We make the following contributions:

1. We show that widely used unlearning methods suffer from shallow unlearning alignment, where spurious unlearning neurons emerge to hide the knowledge rather than erase it.

2. We evaluate this issue in two practical attack scenarios, namely retraining with private data and benign instruction following data, and demonstrate the recoverability of target knowledge, underscoring the need for robust unlearning methods.

3. We introduce SSIUU, a novel method that regularizes the emergence of spurious unlearning neurons and outperforms strong baselines in the two attack scenarios, highlighting its potential for robust and safe deployment of LLM.

## 2 PRELIMINARY: UNLEARNING LANGUAGE MODELS

Typically, unlearning tasks aim to eliminate target knowledge from a language model's parameters, while maintaining other knowledge. Formally, given a language model $P_\theta(y|x) = \prod_{t=1}^{T} P_\theta(y_t|x, y_1, ..., y_{t-1})$ with parameters $\theta$, an unlearning algorithm $g$ updates $\theta$ to $\theta'$, removing target knowledge from $P_\theta$. Various benchmarks consist of input-output pairs $(x, y) \in \mathcal{C}$, where

$\mathcal{C}$ denotes the entire knowledge corpus, which is partitioned into the forget set $\mathcal{C}_f \subset \mathcal{C}$ and the retain set $\mathcal{C}_r \subset \mathcal{C} \backslash \mathcal{C}_f$. Some benchmarks include a test set $\mathcal{C}_t \subset \mathcal{C} \backslash (\mathcal{C}_f \cup \mathcal{C}_r)$ to assess knowledge retention on unseen data. The objective is to train a language model not to output text $y$ when given an input text $x$, where $(x, y) \in \mathcal{C}_f$. However, this training process may destroy the original knowledge of a language model; thus, existing studies have employed the retain set $\mathcal{C}_r$ in the unlearning process to preserve the original knowledge.

## 3 THE SHALLOW UNLEARNING ALIGNMENT PROBLEM

We describe the *shallow unlearning alignment* problem by introducing the concept of *spurious unlearning neuron*, which refers to neurons that acquire new knowledge to suppress the display of target knowledge rather than remove it. To demonstrate this, we show that existing unlearning methods often fail to fully erase the target knowledge, as revealed by their vulnerability to retraining perturbations. Furthermore, our case studies systematically uncover the emergence of spurious unlearning neurons through explainability-driven analyses of model internals.

### 3.1 THE RETRAINING PERTURBATION SCENARIOS

Prior studies demonstrate that fine-tuning a language model often induces catastrophic forgetting (French, 1999; Kirkpatrick et al., 2017) of its original knowledge, and the same phenomenon occurs after the unlearning process (Deeb & Roger, 2024; Hu et al., 2024). This vulnerability is particularly severe when unlearning yields merely shallow alignment, serving as a shortcut rather than a faithful solution. The primary goal of unlearning is to eliminate the target knowledge; however, when alignment is superficial and the target knowledge remains intact, it can easily resurface through subsequent retraining attacks. Therefore, such retraining attack scenarios are not only well-suited for evaluating knowledge retention, but also highly realistic, as many recent LLM platforms provide fine-tuning APIs for customization (e.g., OpenAI) or release their models as open-source (e.g., Meta's Llama and Alibaba's Qwen series).

We assume that users fine-tune an unlearned model with a small fraction ($p$) of instances from the forget set, either intentionally or not. This is called the **harmful retraining attack** setting. If a model undergoes only superficial unlearning, retraining on a subset $p$ of the forget set can lead to the recovery of further forgotten knowledge that is disjoint from the attack dataset. In addition, we consider a benign fine-tuning scenario, in which users retrain an unlearned model with a dataset unrelated to the forget set and without malicious intent. We refer to this as the **benign retraining attack** setting. If forgotten knowledge is nevertheless recovered, it reveals that the unlearned model remains vulnerable to severe security risks.

### 3.2 UNLEARNED MODELS ARE VULNERABLE TO RETRAINING PERTURBATIONS

Based on the scenarios mentioned above, we evaluate widely used unlearning methods to demonstrate their vulnerability: Gradient Ascent (GA), Gradient Difference (GD), Direct Preference Optimization (DPO), NPO, and RMU. The explanation and implementation details of these unlearning methods are described in Appendix C.2. We utilize the FaithUn (Yang et al., 2025) dataset to unlearn a model and to construct datasets for the harmful retraining attack. Specifically, we first unlearn instruction-tuned Llama 3.2 (3B) model with the forget (5%) and retain set (10%) in the dataset using an unlearning method. Then, we fine-tune the unlearned models on a small portion of the forget set $p \in \{0.1, 0.3\}$ and evaluate it using accuracy on the remaining forgotten instances, which are disjoint from the attack dataset. We also utilized 1,000 instances of the Alpaca (Taori et al., 2023) dataset for the benign retrain-

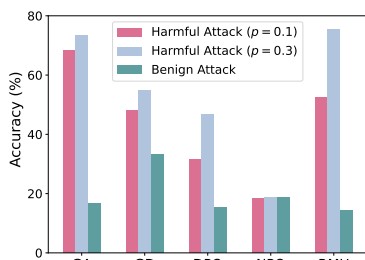

Figure 2: **Experiments on Retraining Attacks with FaithUn.** Their accuracy on the forget set before the attacks is 0%.

ing attack. The details of the attack scenarios are described in Section 6.2.2. Figure 2 shows the vulnerability of unlearned models to the two practical attack scenarios. Our experiments show that unlearned knowledge is substantially recovered through subsequent fine-tuning. For example, with

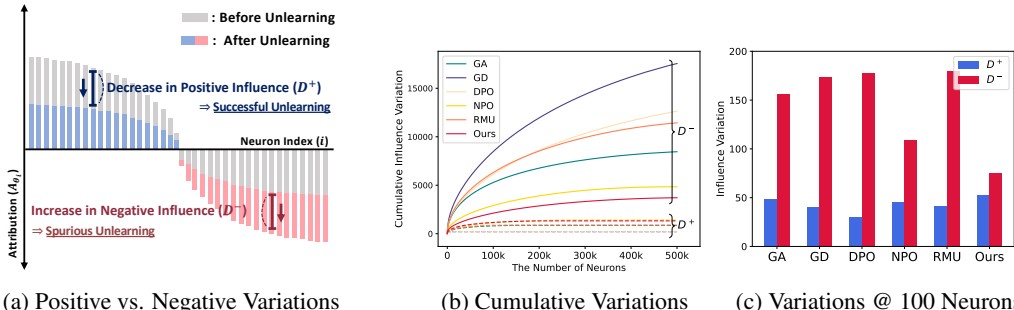

(a) Positive vs. Negative Variations  (b) Cumulative Variations  (c) Variations @ 100 Neurons

Figure 3: **Influence Variations after Unlearning.** After unlearning, most models show that negative influence variations are substantially greater than positive influence variations. In Figure 3-(b), the X-axis denotes the number of accumulated neurons sorted by their scores, and the Y-axis indicates accumulated influence variations. The solid and dotted lines express negative and positive ones, respectively. Figure 3-(c) shows variations extracted from Figure 3-(b) over 100 neurons.

$p = 0.1$ harmful retraining, accuracy in the most significant case exceeds 60. Even under benign retraining, target knowledge is recovered in most unlearned models. These results suggest that current unlearning methods achieve only shallow alignment rather than precisely erasing target knowledge, leaving the unlearning effect fragile.

## 4 THE EXISTENCE OF SPURIOUS UNLEARNING NEURONS

We hypothesize that shallow unlearning alignment arises when neurons evolve to negatively contribute to the target knowledge during unlearning, while the original knowledge-bearing neurons remain intact. Therefore, we define *spurious unlearning neurons* as those that adapt to suppress the output probability of target knowledge after unlearning. To identify such neurons, we employ an attribution method (Yang et al., 2023) to assess whether target knowledge has been genuinely removed from the model's parameters. Specifically, we measure changes in both positive and negative influences of neurons and compare them to determine which direction of influence is predominantly reinforced.

**Quantifying Knowledge.** We use an attribution method (Yang et al., 2023) to quantify the influence of neurons on specific knowledge from a language model. The contribution of an $i$-th neuron to the representation $h$ in a particular layer, in predicting an output text $y$ given an input text $x$ using a language model $P_\theta$, is defined as follows:

$$A_{\theta_i,k}^{(x,y)} = h_{\theta_i,k} \times \frac{\partial P_\theta(y|x)}{\partial h_{\theta_i,k}}, \tag{1}$$

where $h_{\theta_i,k}$ means $k$-th token representation of $i$-th neuron computed by $P_\theta$, and $\partial P_\theta(y|x)/\partial h_{\theta_i,k}$ is the gradient of $P_\theta(y|x)$ with respect to $h_{\theta_i,k}$. We examine transformer variants; thus, the representation and gradient of a specific layer are computed for each input token. Therefore, if an input text includes $K$ tokens, we have $K$ attribution scores for each neuron. In this formula, $h_{\theta_i,k}$ denotes the feature value, and $\partial P_\theta(y|x)/\partial h_{\theta_i,k}$ specifies both the direction and magnitude of how $h_{\theta_i,k}$ contributes to the output probability. Accordingly, we can quantify the contribution of each neuron to the output probability using $A_{\theta_i,k}^{(x,y)}$. If $A_{\theta_i,k}^{(x,y)} > 0$, we can conclude that the $i$-th neuron exerts a positive influence on the output probability. Conversely, if $A_{\theta_i,k}^{(x,y)} < 0$, it exerts negatively. Note that our definition of a neuron, following Yang et al. (2023), refers to each scalar value in the hidden representation $h$. In our analysis, we consider neurons located in the self-attention modules (Q, K, V, and O) as well as in the feed-forward networks (FFNs) across all transformer blocks.

**Quantifying Influence Variations.** We investigate the variations in each direction (i.e., positive and negative) of neuron influence to examine whether target knowledge is effectively erased or not. Specifically, we compute the attribution score before and after unlearning and compare the **decrease in positive influence** and the **increase in negative influence** on the target output to be unlearned.

The decrease in positive influence indicates the desirable result of properly unlearning the target knowledge. However, the increase in negative influence corresponds to the generation of spurious unlearning neurons, which play a role in suppressing the display of target knowledge. The variation of positive influence in each neuron for the whole forget set $\mathcal{C}_f$ is computed as follows:

$$D_i^+ = \frac{1}{n} \times \sum_{(x,y) \in \mathcal{C}_f} \max_k A_{\theta_i,k}^{(x,y)} - \max_k A_{\theta_i',k}^{(x,y)}, \tag{2}$$

where $\theta$ and $\theta'$ are parameters of a model before and after unlearning, respectively. $n$ is the number of samples in the forget set. In contrast, the variation of negative influence $D_i^-$ is similarly computed by altering the *max* aggregation to the *min* aggregation. From this formula, $D_i^+$ quantifies the decrease in positive influence of each neuron, and $D_i^-$ quantifies the increase in the negative influence of each neuron. Figure 3-(a) illustrates the conceptual visualization of the $D_i^+$ and $D_i^-$ scores. Equation 2 is a simplified version of the original equation for better readability. We redefine influence variations as $\tilde{D}_i^+ = \max(D_i^+, 0)$ and $\tilde{D}_i^- = \max(D_i^-, 0)$ since negative values indicate contradictory unlearning behavior. In addition, we replace $A_{\theta_i',k}^{(x,y)}$ with $\max(A_{\theta_i',k}^{(x,y)}, 0)$ in Equation 2, ensuring that over-unlearning does not contribute to the score. Detailed explanations for quantifying influence variations are described in the Appendix A.

**Assessing Whether Unlearning Erases or Hides the Target Knowledge.** We compare the variations in positive and negative influences to uncover evidence of unlearning failure. We sort the neurons independently by $D_i^+$ and $D_i^-$ in decreasing order, select the top-$m$ neurons from each list, and sum their values to compute the aggregated influence variation scores.

Figure 3-(b) shows how the cumulative influence variations increase with the number of neurons across all unlearning methods. Our experimental results on FaithUn indicate that the positive influence variations (dotted lines) are smaller than the negative influence variations (solid lines). This suggests that unlearning methods tend to produce spurious unlearning neurons that merely suppress the display of target knowledge (negative influence variation), rather than faithfully erasing it (positive influence variation).

## 5 METHODOLOGY

To mitigate the emergence of spurious unlearning neurons, we introduce SSIUU, which employs a regularization term to precisely remove target knowledge, instead of suppressing it. Specifically, we optimize the following objective in the unlearning procedure:

$$\arg\min_{\theta^t} \mathcal{L}_{\theta^t} + \lambda \times \sum_{i \in \mathcal{I}^-} \sum_{(x,y) \in C_f} ||A_{\theta_i^{t-1}}^{(x,y)} - A_{\theta_i^t}^{(x,y)}||_2, \tag{3}$$

where $\mathcal{L}_{\theta^t}$ is the loss of an unlearning method (e.g., GA or GD), and $\theta^{t-1}$ and $\theta^t$ are parameters of previous and current optimization steps, respectively. $\mathcal{I}^-$ denotes the set of neuron indices with negative attribution scores for $\theta^t$. The second term of equation 3 measures and minimizes the gap between the attribution of the previous and current steps. This term mitigates the inflation of the negative influence and only reducing the positive influence. The original negative influence (before unlearning) may represent crucial knowledge for language comprehension; therefore, we focus only on avoiding the introduction of additional negative influences while retaining the original negative influences, using the L2-norm as the criterion. For computational efficiency, we derive attribution scores by multiplying each parameter with its gradient, rather than computing attribution scores for every token. We prevent gradients from flowing through $A_{\theta_i^{t-1}}^{(x,y)}$ by treating it as a constant. More details of SSIUU implementation are described in Appendix B.

| Model | Method | FS ($\downarrow$) | RS ($\uparrow$) | US ($\uparrow$) | Harmful Attack ($\downarrow$) $p = 0.1$ | $p = 0.3$ | Benign Attack ($\downarrow$) |
|---|---|---|---|---|---|---|---|
| Llama-3.2 | Default | 91.92 | 89.22 | 57.40 | - | - | - |
| | GA | 0.0 | 58.41 | 54.01 | 68.42 | 73.33 | 16.71 |
| | GD | 0.0 | 81.03 | 55.77 | 48.13 | 54.76 | 33.33 |
| | DPO | 0.0 | 81.47 | 56.90 | 31.58 | 46.67 | 15.34 |
| | NPO | 0.0 | 77.59 | 59.52 | 18.33 | 18.75 | 18.62 |
| | RMU | 0.0 | 77.80 | 56.31 | 52.63 | 75.53 | 14.29 |
| | KLUE | 0.0 | 81.68 | 56.72 | 57.14 | 62.96 | 28.33 |
| | SSIUU | 0.0 | 84.70 | 56.28 | **14.81** | **14.29** | **13.33** |
| Qwen-2.5 | Default | 78.79 | 74.78 | 55.47 | - | - | - |
| | GA | 0.0 | 36.21 | 53.55 | 52.63 | 66.67 | 63.64 |
| | GD | 0.0 | 62.07 | 57.91 | 27.78 | 42.86 | 23.33 |
| | DPO | 0.0 | 60.78 | 53.01 | 47.62 | 58.82 | 36.36 |
| | NPO | 0.0 | 72.63 | 56.69 | 23.33 | 47.91 | 18.18 |
| | RMU | 0.0 | 65.95 | 49.95 | 42.11 | 46.67 | 23.81 |
| | KLUE | 0.0 | 62.50 | 56.94 | 33.33 | 47.92 | 22.73 |
| | SSIUU | 0.0 | 75.86 | 60.01 | **4.76** | **29.41** | **13.04** |

Table 1: **Experimental results on the FaithUn dataset**

# 6 EXPERIMENTS

## 6.1 EXPERIMENTAL SETUPS

**Models and Datasets.** We select Llama-3.2 (3B) (Dubey et al., 2024) and Qwen-2.5 (3B) (Yang et al., 2024a), chosen for their strong NLP performance and broad adoption in prior work. We employ the instruction-tuned version as it more accurately reflects real-world applications of LLMs. We use FaithUn (Yang et al., 2025) and TOFU (Maini et al., 2024) datasets. The main focus of the FaithUn dataset is to erase knowledge of well-known celebrities (e.g., "Which country is Donald Trump from?"). It does not require any fine-tuning before the unlearning process, as real-world knowledge is inherently encoded in language models. Therefore, we use the FaithUn dataset as the primary dataset for analysis, as it represents a practical setting for unlearning. We also utilize the TOFU dataset, a synthetic author profile dataset, to demonstrate that our findings are generalizable. Since TOFU encompasses knowledge of synthetic entities, it requires an additional fine-tuning process before unlearning a language model. Details are in Appendix C.1.

**Baselines.** We utilize widely-used unlearning methods to assess the shallow unlearning alignment: Gradient Ascent (GA) (Jang et al., 2022), Gradient Difference (GD) (Maini et al., 2024), Direct Preference Optimization (DPO) (Rafailov et al., 2023), NPO (Zhang et al., 2024), RMU (Li et al., 2024), and KLUE (Yang et al., 2025). We configure hyperparameter settings by referring to prior studies (Zhang et al., 2024; Jin et al., 2024; Li et al., 2024; Yang et al., 2025). To implement SSIUU, we adopt GD as the backbone unlearning algorithm for computing $\mathcal{L}_{\theta^t}$ in Equation 3, as it is a representative unlearning approach that is simple and widely applicable. Appendix C.2 provides more details on the baselines and our method.

**Training setups.** We finish the unlearning process when a model's scores (Accuracy or ROUGE) for the forget set reach the pre-defined threshold, following the FaithUn benchmark. In the case of FaithUn, we early stop the training procedure when accuracy for the forget set reaches 0.33 (random sampling from three options) to select the optimal model, as it employs Multiple Choice QA (MCQA) settings. For TOFU, we fine-tune the model on the dataset with a learning rate of $\eta = 1e-5$ for 5 epochs. We then construct mismatched QA pairs by randomly pairing questions with incorrect answers, and compute the mean ROUGE-L recall, which defines the unlearning threshold $\tau$. This yields $\tau = 0.1971$, which we use for early stopping in the unlearning process. Note that high learning rates often destroy other knowledge of LLMs during the unlearning process; therefore, we search for the lowest learning rate that still achieves convergence to the threshold.

| Model | Method | FS ($\downarrow$) | RS ($\uparrow$) | US ($\uparrow$) | Harmful Attack ($\downarrow$) $p = 0.1$ | $p = 0.3$ | Benign Attack ($\downarrow$) |
|-------|--------|------|------|------|------|------|------|
| | Default | 90.42 | 93.54 | 91.45 | - | - | - |
| | GA | 19.49 | 87.51 | 91.88 | 48.72 | 54.21 | 23.35 |
| | GD | 19.32 | 89.36 | 92.88 | 45.41 | 51.73 | 20.98 |
| Llama-3.2 | DPO | 19.57 | 92.28 | 91.95 | 87.39 | 88.69 | 26.60 |
| | RMU | 14.04 | 93.70 | 91.74 | 87.97 | 90.97 | **20.33** |
| | KLUE | 18.71 | 86.86 | 92.81 | 43.71 | 49.38 | 22.19 |
| | Sᴄɪᴜᴜ | 17.75 | 92.74 | 91.67 | **31.82** | **37.53** | 21.08 |

Table 2: **Experimental results on the TOFU dataset**[*]

## 6.2 Evaluation Metrics

### 6.2.1 Basic unlearning metrics.

To implement the basic unlearning framework, we follow the original implementation of the FaithUn and TOFU datasets. We adopt three metrics (FS, RS, and US) to show the results after unlearning. We utilize accuracy and ROUGE-L recall, a measure of sentence structure similarity (Lin, 2004), to compute those scores for the FaithUn and TOFU datasets, respectively. **(1) Forgetting Score (FS)**: This score is computed for the forget set to evaluate the basic unlearning performance. We use 5% and 1% of data samples as the forget set for FaithUn and TOFU, respectively. We do not consider the overwhelming forget set scenario, as our study begins with unlearned models that appear effective. **(2) Retention Score (RS)**: This score is used to evaluate the retention performance of other knowledge. It is computed using data instances that are disjoint from the forget set. We employ the pre-specified retention sets provided in FaithUn and TOFU. **(3) Utility Score (US)**: This score is used to evaluate the retention of general utility knowledge. We use five language comprehension datasets—MMLU (Hendrycks et al., 2020), GSM8K (Cobbe et al., 2021), Hellaswag (Zellers et al., 2019), ARC-Challenge (Clark et al., 2018), and Winogrande (Sakaguchi et al., 2021)—to evaluate the general language understanding capabilities of unlearned models on FaithUn. We sample 500 instances to assess models on each dataset. For TOFU, we adopt the utility datasets provided by the benchmark, namely Real Author and World Facts. Further details are provided in Appendix C.1.

### 6.2.2 Unlearning robustness metrics.

We propose two attack scenarios to evaluate the robustness of unlearning methods. **(1) Harmful Attack Score**: We retrain an unlearned model on a small portion $p \in \{0.1, 0.3\}$ of the forget set with learning rates $\eta \in \{10^{-5}, 5 \times 10^{-6}, 10^{-6}\}$, and report the maximum Forgetting Score (FS) observed during retraining on the remainder of the forget set disjoint from the retraining data. A significant recovery of FS after the attack indicates that the target knowledge has not been fully removed. For FaithUn, we construct the retraining attack pool using only falsely answered instances, as its MCQA evaluation framework that finalizes the unlearned model when FS $\leq 0.33$. Thus, instances correctly answered by chance are excluded from the forget set in privacy attack scenarios. **(2) Benign Attack Score**: We retrain an unlearned model using the Alpaca dataset, which is an instruction-following benchmark. We use 1,000 data instances of the dataset to retrain an unlearned model with varying learning rates $\eta \in \{10^{-5}, 10^{-6}, 10^{-7}\}$, and report the maximum Forgetting Score (FS) in the attack process. The basis for selecting the learning rate search spaces is provided in Appendix D.2. We run each attack scenario three times and report the mean scores.

### 6.3 Sᴄɪᴜᴜ improves the robustness of unlearning

We conduct experiments on the FaithUn and TOFU datasets to demonstrate that our method substantially improves the robustness of unlearning against retraining attacks. Table 1 shows the experimental results in the FaithUn dataset, and Table 2 shows the experimental results in the TOFU dataset. First, we conclude that most unlearning processes are effective, as the Retention Scores (RS) and Utility Scores (US) remain close to the default after unlearning. However, unlearned models ex-

---

[*]We exclude the NPO results on TOFU, as the model fails to converge and tends to collapse after unlearning.

hibit fragility under both attack scenarios. Our method outperforms others by exhibiting robustness against these attacks, demonstrating that target knowledge is significantly removed.

Furthermore, existing unlearning methods (e.g., GD) can inadvertently highlight target knowledge due to excessive variations in negative influence during removal. This over-unlearning renders models vulnerable to membership inference attacks (MIAs) (Jin et al., 2024; Di et al., 2024). Following Yang et al. (2024b), we confirm this phenomenon using the logit lens, which is a tool to interpret intermediate activations in transformer models, as shown in Figure 4. Specifically, we project intermediate representations through the word embedding layer and derive the probability of the golden answer and a randomly chosen distractor. By comparing these probabilities, we compute the accuracy of the golden answer (binary classification) to determine whether the representation retains the knowledge. As a result, the accuracy is far below ($\approx 0.2$) the chance level (0.5) for GD, particularly in layers 18–27 (Figure 4-a), indicating a failure to achieve robust or faithful unlearning. In contrast, SsIUUachieves chance-level accuracy after unlearning (Figure 4-b), demonstrating robust unlearning at the inner-layer representational level by suppressing spurious unlearning neurons.

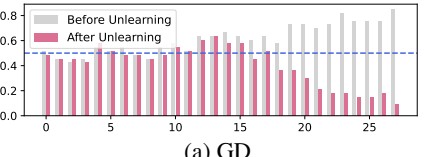 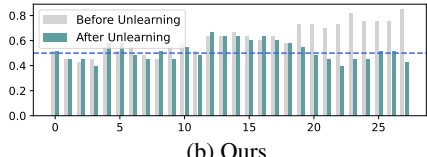

(a) GD                (b) Ours

Figure 4: **Analyzing Excessive Knowledge Removal via Logit Lens.** The X-axis and Y-axis correspond to layer indices and accuracy, respectively. The blue dotted line represents the random-choice baseline (binary classification). GD tends to excessively unlearn target knowledge, whereas SsIUU adequately unlearns it to the random-choice level.

### 6.4 SsIUU MITIGATES THE EMERGENCE OF SPURIOUS UNLEARNING NEURONS

To examine the suppression of spurious unlearning neurons, we analyze influence variations across modules and layers on Llama-3.2 using FaithUn. Figure 5 compares GD and SsIUU, showing that SsIUU mitigates the spurious unlearning neurons by reducing negative influence variations. In GD, target knowledge is mainly removed in the last layers, as positive influence variation appears only there. In contrast, with SsIUU, positive influence variation emerges in a broad range of modules, indicating that knowledge distributed in multiple modules is effectively removed. Moreover, GD shows a strong increase in negative influence variation, whereas SsIUU demonstrates a clear suppression of such effects. The results also indicate that spurious unlearning neurons primarily emerge in the Attention Q and K modules, where they cut the knowledge connections between tokens.

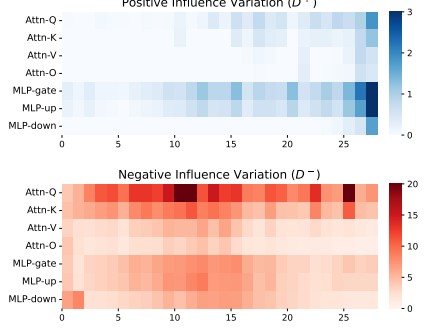 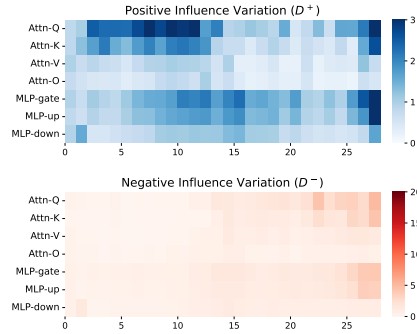

(a) Influence Variation Results of GD    (b) Influence Variation Results of Ours

Figure 5: **Influence Variation for Each Module and Layer.** We plot positive and negative influence variations of GD and SsIUU for each module and layer. X-axis and Y-axis correspond to layer indices and module type, respectively. The color scale indicates the average variation in influence for the top-100 neurons in each module.

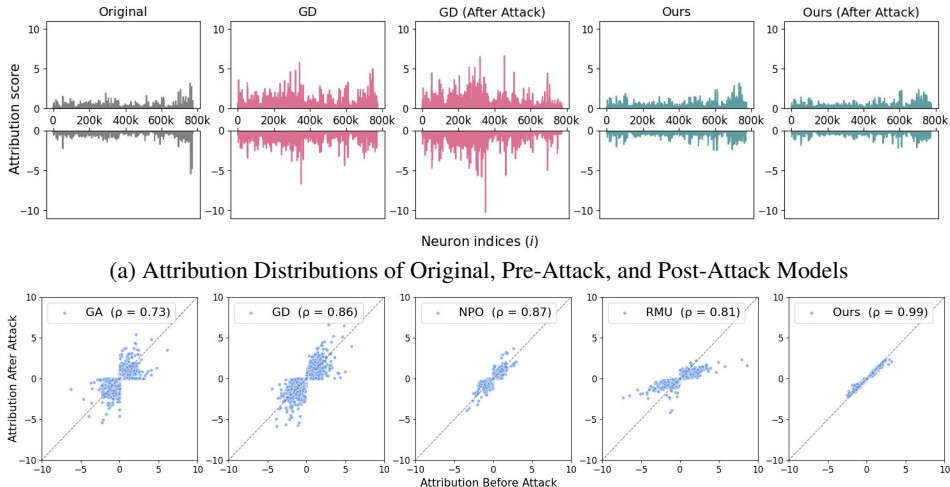

(a) Attribution Distributions of Original, Pre-Attack, and Post-Attack Models

(b) Correlation ($\rho$) of Attribution Distributions between Pre- and Post-Attack Models

Figure 6: **Deeper Investigations into Influence Distributions.** We present the influence (attribution) changes after the harmful attack ($p = 0.1$). Figure 6-(a) illustrates the attributions of the original model, unlearned models, and models after the attack. Figure 6-(b) presents the correlation between attributions before and after the attacks. While models trained with other methods exhibit high variability, our method yields relatively consistent distributions with a strong correlation.

## 6.5 RETRAINING ATTACK DISRUPTS THE SPURIOUS UNLEARNING NEURONS

To demonstrate that the attribution space is a key factor in robust unlearning, we analyze attribution spaces before and after the attack on Llama-3.2 using FaithUn. Figure 6-(a) presents the attribution distributions of the original Llama-3.2, unlearned models with GD and SSIUU, and models retrained with attacks. In Figure 6-(a), GD produces numerous spurious unlearning neurons with strong negative attributions. Surprisingly, the positive attributions also increase, indicating that unlearning induces a competitive emergence in both directions of contribution, often within the same layer. This competition becomes even more pronounced after the attack, resulting in high variability across the distributions. In contrast, SSIUU suppresses this bidirectional competition and maintains distributional consistency, demonstrating robustness even under attack.

Furthermore, we investigate the correlation of the attribution distributions between before and after the harmful retraining attack ($p = 0.1$), as shown in Figure 6-(b). In Table 1, GA shows significant vulnerability, and NPO shows relatively lower vulnerability to the attacks. These results are also supported by Figure 6-(b), demonstrating that GA shows a low correlation score ($\rho = 0.73$) and NPO shows a relatively high correlation score ($\rho = 0.87$) in attribution distributions between before and after the attack. Note that while GD and NPO exhibit similar correlations, the variance of NPO is substantially lower than that of GD. Our method yields the highest correlation score ($\rho = 0.99$) compared to other methods, indicating the greatest stability against the attack.

## 7 RELATED WORKS

**Machine Unlearning for LLMs.** Unlearning has been proposed as a method to tackle privacy and copyright concerns in LLM text generation (Jang et al., 2022; Yao et al., 2023; Barbulescu & Triantafillou, 2024; Yao et al., 2024). Notable studies have explored it via gradient ascent (Jang et al., 2022; Maini et al., 2024), preference optimization (Jin et al., 2024; Yang et al., 2025), and representation learning (Li et al., 2024). Prior studies have assessed the capability of these methods by introducing new benchmarks in various domains. WHP and MUSE (Eldan & Russinovich, 2023; Shi et al., 2024) have used copyrighted texts. TOFU (Maini et al., 2024) has built synthetic author data to unlearn. WMDP (Li et al., 2024) has dealt with mitigating hazardous knowledge in professional domains, such as biosecurity. RWKU and FaithUn (Jin et al., 2024; Yang et al., 2025) have examined real-world entity knowledge, evaluating generalization across various prompt designs. These efforts have laid the groundwork for assessing unlearning algorithms. However, their attempts to interpret and analyze their behavior to address the underlying issues remain limited.

**Unlearning Robustness.** Prior studies have shown that the alignment of knowledge in LLMs can be easily compromised by additional training (Qi et al., 2023; 2024; Huang et al., 2024a;b). A similar phenomenon has also been observed in the context of unlearning. Several studies have attempted to demonstrate the robustness of unlearning against retraining attacks (Deeb & Roger, 2024; Hu et al., 2024). They formulated an unlearning–retraining pipeline that incorporates both harmful and benign threat models, and revealed that unlearning is also fragile. However, these works have not investigated the underlying cause of unlearning fragility. In this paper, we take a first step toward unveiling one of its key reasons: existing unlearning algorithms tend to hide knowledge rather than erase it, as revealed through an explainability-based analysis.

## 8 CONCLUSION

We demonstrated using attribution-based analysis that existing unlearning methods induce spurious unlearning neurons, which hide target knowledge rather than erase it. Our experiments in two realistic attack scenarios, retraining with private and benign datasets, further reveal that such a shallow alignment leaves models vulnerable to knowledge recovery, undermining their reliability in practice. To overcome these shortcomings, we propose SSIUU, a method to regularize the growth of negative influence to suppress the emergence of spurious unlearning neurons. Our results confirm that SSIUU faithfully removes target knowledge and maintains robustness against subsequent retraining. By more directly aligning unlearning with the removal of target knowledge, our approach represents a promising advance toward more reliable LLM deployment.

## ACKNOWLEDGEMENT

N. Yang and K. Jung were also supported by Institute of Information & communications Technology Planning & Evaluation (IITP) grant funded by the Korea government (MSIT) [No.RS-2022-II220184, Development and Study of AI Technologies to Inexpensively Conform to Evolving Policy on Ethics & No.RS-2021-II211343, Artificial Intelligence Graduate School Program (Seoul National University)] and the Ministry of Education of the Republic of Korea and the National Research Foundation of Korea (NRF-2024S1A5C3A01042642).

## ETHICS STATEMENT

This paper highlights the vulnerability of unlearned models to retraining attacks and introduces a novel method to achieve robust unlearning. While our framework may inadvertently imply attack strategies that malicious users could exploit in fine-tuning API platforms or open-sourced models, we believe that openly addressing these risks will ultimately foster safer and more reliable deployment of language models.

## REPRODUCIBILITY STATEMENT

We have made significant efforts to ensure the reproducibility of our work. The details of the FaithUn and TOFU datasets for their utilization are provided in Appendix C.1. The specifications of our method and the baseline approaches are described in Appendix C.2. Furthermore, the training procedure is outlined in Section 6.1 and Appendix C.1. Furthermore, all experimental settings and results are described in detail within the respective sections.

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

## A    DETAILS OF INFLUENCE VARIATION QUANTIFICATION

The primary aim of our work is to reveal one of the key reasons for unlearning failure: the emergence of spurious unlearning neurons, which increase only negative influences (hiding knowledge), rather than decreasing existing positive influences (erasing knowledge). In this section, we provide details on the quantification of influence variation. Using Equation 1, we quantify the contribution of each neuron to predicting the probability of the output $y$ given the input $x$. Each neuron has its own contribution value, and since activations and gradients are computed for each token representation, numerous attribution values are obtained even for a single neuron and a single fact. Furthermore, contributions may vary depending on the input tokens processed or the output positions predicted, even within one $(x, y)$ pair. Therefore, a robust aggregation strategy is required to quantify each neuron's contribution to knowledge. An existing work (Yang et al., 2023) has shown that *max aggregation* over all tokens achieves higher performance than *mean aggregation*. This finding is analogous to the use of *max pooling* in Convolutional Neural Networks (CNNs), where selecting the maximum values in a feature map captures the most salient information, often leading to better performance than *mean pooling*. Motivated by these insights, we adopt *max aggregation* and *min aggregation* to capture salient positive and negative influences, respectively, as follows:

$$D_i^+ = \frac{1}{n} \times \sum_{(x,y) \in \mathcal{C}_f} \max_k A_{\theta_i, k}^{(x,y)} - \max_k A_{\theta_i', k}^{(x,y)}, \tag{4}$$

$$D_i^- = \frac{1}{n} \times \sum_{(x,y) \in \mathcal{C}_f} \min_k A_{\theta_i, k}^{(x,y)} - \min_k A_{\theta_i', k}^{(x,y)}, \tag{5}$$

where $\theta$ and $\theta'$ are parameters of a model before and after unlearning, respectively. $n$ is the number of samples in the forget set. From these equations, we can quantify each neuron's influence variation in the positive and negative directions. $D_i^+$ measures the positive influence variation of the $i$-th neuron. If the original attribution value is positive and this value is reduced after unlearning, the result can be interpreted as a desirable outcome, since the existing positive contribution to the output probability is diminished. However, if the original attribution value is negative and this value becomes further reinforced after unlearning, it can be regarded as an undesirable outcome, as the unlearning process induces a spurious unlearning neuron that contributes only negatively. If only the negative contributions are reinforced, then the original positive contributions may remain intact. Therefore, we measure and compare these two metrics ($D^+$ and $D^-$) to evaluate the faithfulness of unlearning behaviors.

However, Equation 4 and 5 do not account for two undesirable unlearning results: *contradictory unlearning behavior* and *over-unlearning*. Contradictory unlearning behavior refers to cases where neuronal contributions vary in a way that increases the output probability of target knowledge. As discussed in Section 6.5, unlearning methods often increase both positive and negative neuronal contributions in a competitive manner. Thus, although inherently contradictory, this behavior is frequently observed during unlearning. Since it falls outside the scope of our focus, we redefine the influence variations as $\tilde{D}_i^+ = \max(D_i^+, 0)$ and $\tilde{D}_i^- = \max(D_i^-, 0)$ to exclude the effects of contradictory unlearning behavior in computing the final scores. Equation 4 also fails to filter out over-unlearning. For example, if the original attribution is positive but becomes negative after unlearning, the $D^+$ score may increase substantially. As noted in Section 6.3, this represents undesirable unlearning, which makes models vulnerable to membership inference attacks (MIAs). To address this, we replace $A_{\theta_i', k}^{(x,y)}$ with $\max(A_{\theta_i', k}^{(x,y)}, 0)$ in Equation 4, ensuring that over-unlearning does not contribute to the score.

## B    DETAILS OF SSIUU IMPLEMENTATION

This section describes the details of SSIUU implementation. SSIUU algorithm computes the attribution of each neuron, and suppresses the inflation of the negative influence variation ($D^-$). SSIUU includes the minimization problem of the following objective:

$$\mathcal{J}(\theta^t) = L_{\theta^t} + \lambda \times \sum_{i \in \mathcal{I}^-} \sum_{(x,y) \in C_f} (A_{\theta_i^{t-1}}^{(x,y)} - A_{\theta_i^t}^{(x,y)})^2, \tag{6}$$

where $A_{\theta_i^{t-1}}^{(x,y)}$ is treated as a constant since the parameters at the previous step are fixed. Therefore, we derive the final gradient of SSIUU as follows:

$$\Delta_i^{(x,y)}(\theta^t) := A_{\theta_i^t}^{(x,y)} - A_{\theta_i^{t-1}}^{(x,y)} \tag{7}$$

$$\mathcal{J}(\theta^t) = L_{\theta^t} + \lambda \times \sum_{i \in \mathcal{I}^-} \sum_{(x,y) \in C_f} (\Delta_i^{(x,y)}(\theta^t))^2 \tag{8}$$

$$\nabla_{\theta^t} \mathcal{J}(\theta^t) = \nabla_{\theta^t} L_{\theta^t} + \lambda \sum_{i \in \mathcal{I}^-} \sum_{(x,y) \in C_f} \nabla_{\theta^t} (\Delta_i^{(x,y)}(\theta^t))^2 \tag{9}$$

$$\nabla_{\theta^t} (\Delta_i^{(x,y)}(\theta^t))^2 = 2\Delta_i^{(x,y)}(\theta^t) \nabla_{\theta^t} \Delta_i^{(x,y)}(\theta^t) = 2\Delta_i^{(x,y)}(\theta^t) \nabla_{\theta^t} A_{\theta_i^t}^{(x,y)} \tag{10}$$

$$\nabla_{\theta^t} \mathcal{J}(\theta^t) = \nabla_{\theta^t} L_{\theta^t} + \lambda \sum_{i \in \mathcal{I}^-} \sum_{(x,y) \in C_f} 2\Delta_i^{(x,y)}(\theta^t) \nabla_{\theta^t} A_{\theta_i^t}^{(x,y)} \tag{11}$$

We now proceed to discuss the computation of $\nabla_{\theta^t} A_{\theta_i^t}^{(x,y)}$. For computational efficiency, the SSIUU algorithm in this work treats each parameter as a feature, rather than using the neuron representation $h$. Accordingly, for each scalar parameter $\phi_{t,i}$, the attribution is computed as follows:

$$A_{\phi_{t,i}}^{(x,y)} = \phi_{t,i} \cdot g_{t,i}^{(x,y)}; \qquad g_{t,i}^{(x,y)} := \frac{\partial P_{\phi_t}(y|x)}{\partial \phi_{t,i}}, \tag{12}$$

where $P_{\phi_t}$ is the identical to $P_{\theta^t}$. Note that in the original attribution formulation (Equations 1 and 2), a neuron was defined as a single value within the representation computed at each layer; thus, each column of the weight matrix corresponded to the $i$-th neuron. In contrast, in the SSIUU regularization, attribution values are computed for each scalar element within the parameters themselves. Then the computation of $\nabla_{\phi_t} A_{\phi_{t,i}}^{(x,y)}$ is as follows:

$$\frac{\partial A_{\phi_{t,i}}^{(x,y)}}{\partial \phi_{t,j}} = \frac{\partial (\phi_{t,i} g_{t,i}^{(x,y)})}{\partial \phi_{t,j}} = \frac{\partial \phi_{t,i}}{\partial \phi_{t,j}} g_{t,i}^{(x,y)} + \phi_{t,i} \frac{\partial g_{t,i}^{(x,y)}}{\partial \phi_{t,j}} = \delta_{i,j} g_{t,i}^{(x,y)} + \phi_{t,i} \cdot \frac{\partial g_{t,i}^{(x,y)}}{\partial \phi_{t,j}}, \tag{13}$$

where $\delta_{i,j}$ denotes the Kronecker delta, and since $\frac{\partial g_{t,i}^{(x,y)}}{\partial \phi_{t,j}} = \frac{\partial^2 P_{\phi_t}(y|x)}{\partial \phi_{t,i} \partial \phi_{t,j}}$, the overall gradient of SSIUU becomes:

$$\frac{\partial \mathcal{J}(\phi_t)}{\partial \phi_{t,j}} = \frac{\partial L_{\phi_t}}{\partial \phi_{t,j}} + 2\lambda \sum_{(x,y) \in C_f} \sum_{i \in \mathcal{I}^-} \Delta_i^{(x,y)}(\phi_t)[\delta_{i,j} g_{t,i}^{(x,y)} + \phi_{t,i} \cdot \frac{\partial^2 P_{\phi_t}(y|x)}{\partial \phi_{t,i} \partial \phi_{t,j}}], \tag{14}$$

where $\mathcal{I}^-$ denotes the set of globally selected indices at iteration $t$, computed over the entire forgetting dataset. Specifically, we obtain $\mathcal{I}^-$ by identifying parameters whose aggregated attribution values $A_{\phi_{t,i}} := \sum_{(x,y) \in C_f} (\phi_{t,i} \cdot \partial P_{\phi_t}(y|x) / \partial \phi_{t,i})$ are negative. In practice, we approximate this aggregation using the current forget batch.

In the Hessian term of the final gradient, if we define $v_i^{(x,y)} := 2\lambda \Delta_i^{(x,y)}(\phi_t) \phi_{t,i}$ and $H^{(x,y)} := \nabla_{\phi_t}^2 P_{\phi_t}(y|x) \in \mathbb{R}^{d \times d}$, then the second derivative term in Equation 14 can be written as $\left( H^{(x,y)} v^{(x,y)} \right)_j$ for each $(x,y)$ pair. This is efficiently computed via Hessian-vector product (HVP)

---

**Algorithm 1** SSIUU: Suppressing Spurious Unlearning Neurons for Robust Unlearning

---

**Input:** Regularization weight $\lambda$; Learning rate $\eta$; Optimization steps $T$
**Output:** Unlearned model $P_{\phi_T}$

1: **for** $t = 1$ to $T$ **do**:
2:     Sample a batch $b$ of forget data $(x, y) \in C_f$
3:     Sample a batch $b'$ of retain data $(x', y') \in C_r$
4:     Compute GD loss $\mathcal{L}_{GD}$ using $b$ and $b'$
5:     $A_{\phi_{t,i}} \leftarrow \sum_{(x,y) \in b} \left( \phi_{t,i} \cdot \partial P_{\phi_t}(y|x)/\partial \phi_{t,i} \right)$  // parameter-level attribution
6:
7:     **if** $t = 1$ **then**:
8:         $\mathcal{L}_{\text{SSIUU}} \leftarrow 0$
9:     **else**:
10:         $\mathcal{I}^- \leftarrow \{i | A_{\phi_{t,i}} < 0\}$
11:         $\mathcal{L}_{\text{SSIUU}} \leftarrow \sum_{i \in \mathcal{I}^-} (A_{\phi_{t-1,i}} - A_{\phi_{t,i}})^2$
12:     $A_{\phi_{t-1,i}} \leftarrow A_{\phi_{t,i}}$   // detach from the computation graph
13:     $\hat{\mathcal{L}} = \mathcal{L}_{GD} + \lambda \mathcal{L}_{\text{SSIUU}}$
14:     $\phi_t \leftarrow \phi_t - \eta \nabla_{\phi_t} \hat{\mathcal{L}}$
15: **end for**

---

using $H^{(x,y)} v^{(x,y)} = \nabla_{\phi_t}((\nabla_{\phi_t} P_{\phi_t}(y|x))^\top v^{(x,y)})$, without constructing the full Hessian. Therefore, the additional computational overhead is of the same order as a standard gradient computation. Algorithm 1 shows the gradient computation process of SSIUU.

## C    DETAILS IN EXPERIMENTS

### C.1    DATASET DETAILS

We use FaithUn (Yang et al., 2025) and TOFU (Maini et al., 2024) datasets for the experiments.

**FaithUn.**    The main focus of FaithUn is to erase knowledge of well-known 200 celebrities. FaithUn includes 664 data instances, and we follow all the unlearning setups of the original paper. Specifically, we select 5%, 10%, and 70% of the dataset for the forget set, retain set, and test set, respectively. We use the forget set and retain set in the unlearning process, and report the accuracy for the forget set and test set in Table 1. In the unlearning process, we utilize just question-answer pairs; in contrast, we use an MCQA template for evaluating unlearned models. The MCQA template includes an instruction, a question, and answer options, as shown in Table 3. In unlearning, there is a trade-off between forgetting knowledge and retaining overall knowledge, making it difficult to select the optimal model. For fair comparison, we early stop when Forget Score (FS) $\leq 0.33$ to choose the model (the random-choice level for three options), following the original paper.

> Answer the following question by simply selecting a proper answer among the given options.  You must generate only the exact word without an explanation.\nQuestion: {$question}\nOptions: {$options}\nYour Answer:

Table 3: The Multiple Choice QA Evaluation Template for FaithUn

**TOFU.**    The focus of the TOFU dataset is to erase knowledge of synthetic QA instances. Therefore, we must fine-tune LLMs with the TOFU dataset before unlearning to make LLMs have knowledge about the dataset. The TOFU dataset includes four types of datasets: Forget set, Retain set, Real Author, and World Facts datasets. The Forget set is the basic dataset used for forgetting. The Retain set is used to evaluate the retention ability of TOFU knowledge. The Real Author and World Facts datasets are used to evaluate the retention of other knowledge.  TOFU includes a total of 4,000 instances, and we utilize a pre-defined 1% Forget set for unlearning. We finetune the model on the dataset ($\eta = 1e - 5$ and 5 epochs) to have knowledge about TOFU, following the original paper.

To determine the early stopping threshold, we first construct mismatched QA pairs by randomly selecting incorrect answers and compute their mean ROUGE-L recall to set the unlearning threshold $\tau$. We finally set $\tau = 0.1971$ and use it for early stopping in the unlearning process.

## C.2 BASELINE DETAILS

We adopt a range of widely used unlearning methods to evaluate shallow unlearning alignment: Gradient Ascent (GA) (Jang et al., 2022), Gradient Difference (GD) (Maini et al., 2024), Direct Preference Optimization (DPO) (Rafailov et al., 2023), Negative Preference Optimization (NPO) (Zhang et al., 2024), RMU (Li et al., 2024), and Knowledge-Localized Unlearning (KLUE) (Yang et al., 2025). Hyperparameter settings are selected in reference to prior work (Zhang et al., 2024; Jin et al., 2024; Li et al., 2024; Yang et al., 2025).

**(1) Gradient Ascent (GA)**: Unlike gradient descent in pre-training, GA maximizes the negative log-likelihood loss on the forget set, driving the model away from its original predictions.

**(2) Gradient Difference (GD)**: Since GA may also erase unrelated knowledge, GD incorporates an auxiliary retention loss to maximize the likelihood of the retain set, preserving irrelevant knowledge.

**(3) Direct Preference Optimization (DPO)**: DPO applies preference optimization for unlearning by contrasting positive and negative instances. We treat the correct answer as the negative instance and use a rejection response ("I can't answer the question.") as the positive instance, along with an auxiliary retention loss to maintain logits on the retain set.

**(4) Negative Preference Optimization (NPO)**: NPO extends DPO by discarding positive examples. We implement DPO and NPO with an auxiliary retention loss and search $\beta \in [0.1, 0.5]$.

**(5) RMU**: RMU randomizes intermediate representations for the forget set. We search $\alpha \in \{20, 50, 100, 150, 200, 300\}$ and use $c = 20$ and $l = 7$, following the original implementation[†].

**(6) KLUE**: KLUE localizes target knowledge by identifying and updating only a small subset of neurons. We update 10% of neurons in feed-forward networks using GD. We apply Superficial Knowledge Regularization with $\alpha = 10$ and $N = 5$, following the original implementation.

**(7) SSIUU**: Our method builds on GD as the backbone unlearning algorithm to compute $\mathcal{L}_{\theta_t}$ (Equation 3), given its simplicity and broad applicability. We set $\lambda = 0.001$ for FaithUn and $\lambda = 0.0001$ for TOFU. The basis for selecting $\lambda$ is provided in Appendix D.1.

## D ADDITIONAL EXPERIMENTAL RESULTS

### D.1 EXPERIMENTS ON THE REGULARIZATION WEIGHT TERM ($\lambda$)

We conduct experiments to select an appropriate weight for the regularization term in Equation 3, as shown in Figure 7-(a). We search over $\lambda \in \{0.0001, 0.001, 0.01, 1.0\}$ and evaluate unlearned models on the FaithUn dataset. We report accuracy on (i) the retain set ($C_r$), used during unlearning, and (ii) the test set ($C_t$), which is unseen. These two accuracies should be maintained, as they indicate retention performance. We also measure robustness by computing accuracy after attacks ($p = 0.1$ and $p = 0.3$). With a small weight ($\lambda = 0.0001$), the unlearned model is highly vulnerable to both attacks. Increasing the weight ($\lambda = 0.001$ or $\lambda = 0.01$) mitigates this vulnerability, but retention performance starts to drop when $\lambda \geq 0.01$. Specifically, at $\lambda = 0.01$, accuracy on the test set degrades, and at $\lambda = 1.0$, accuracy on the retain set also drops, indicating that the model fails to converge properly. These results highlight the necessity of carefully selecting the regularization weight.

### D.2 EXPERIMENTS ON THE ATTACK LEARNING RATES

We search varying learning rates for the attacks, as shown in Figure 7-(b) and (c). We reveal that the harmful and benign attacks are fatal in the range of $[10^{-5}, 10^{-6}]$ and $[10^{-6}, 10^{-7}]$, respectively. From these results, we decide to search learning rates $\eta = \{10^{-5}, 5 \times 10^{-6}, 10^{-6}\}$ and $\eta = \{10^{-5}, 10^{-6}, 10^{-7}\}$ for the harmful and the benign attacks.

---

[†]https://github.com/centerforaisafety/wmdp

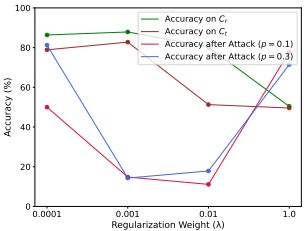 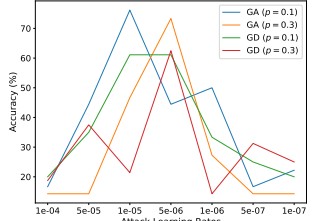 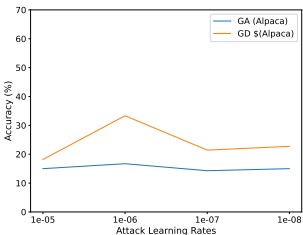

(a) The Effect of Regularization Weight ($\lambda$) in SSIUU

(b) The Learning Rates Experiments on the Harmful Attacks

(c) The Learning Rates Experiments on the Benign Attack

Figure 7: **Additional Experiments.** Figure 7-(a) shows the accuracy for varying regularization weight ($\lambda$). We compute the accuracy on the retain set and test set to assess the models' behavior after unlearning. We also report the accuracy after the harmful attacks. Figure 7-(b) and (c) show the experimental results for varying learning rate attacks on GA and GD models.

# E  LIMITATIONS AND DISCUSSION

By analyzing variations in positive and negative neuronal influence, we find that current unlearning methods often exhibit shallow unlearning alignment. We also discuss future directions and the practical challenges associated with our approach.

**The Challenge of Post-Unlearning Stability.** The retraining attack scenarios we examine reflect realistic deployment environments, especially as fine-tuning APIs and open-source checkpoints become increasingly accessible. Our findings show that shallow unlearning alignment can make forgotten knowledge more recoverable, even through benign forms of instruction tuning. This indicates that secure unlearning is not solely a model-level challenge but also a pipeline-level one: platform operators must account for how downstream fine-tuning, user-supplied data, and model-editing workflows may reintroduce or reconstruct removed information.

**Toward Advanced Knowledge Quantification for Dynamic Interpretability.** Our analysis introduces a new research direction that leverages interpretability methods to track how knowledge representations change before and after unlearning, thereby enabling a more faithful evaluation of unlearning behavior in dynamic settings. While our findings are consistent across multiple unlearning methods and LLM architectures, they also reveal natural methodological dependencies, as our analysis relies on the attribution-based method proposed by Yang et al. (2023). Existing knowledge quantification approaches (Wang et al., 2022; Panigrahi et al., 2023; Zhu et al., 2025; Zhao et al., 2025) are not designed to support neuron-level negative attribution, limiting their applicability to our objective. This suggests that further work is needed to develop knowledge quantification techniques capable of supporting dynamic, attribution-level interpretability for unlearning—moving beyond simple positive-correlation measures toward more sophisticated, mathematically grounded approaches.

**Improving Scalability and Adaptivity of SSIUU.** SSIUU introduces a principled way to mitigate the emergence of spurious unlearning neurons, but it also raises practical considerations. The regularization term constrains internal attribution dynamics at every optimization step, which may pose scalability challenges for extremely large models or very long sequences. Although the Hessian–vector product computation keeps the overhead manageable, exploring more efficient approximations—such as sparse or layer-specific variants—could further broaden its applicability. In our experiments, we selected the regularization weight ($\lambda$) after a simple hyperparameter search, and we found that a single value generalized across multiple models. Nonetheless, careful tuning can further improve its performance. Future work could incorporate more adaptive strategies, such as dynamically adjusting $\lambda$, to further improve usability and stability.

