# OpenReview forum: "Erase or Hide? Suppressing Spurious Unlearning Neurons for Robust Unlearning"
_ICLR.cc/2026/Conference — ICLR 2026 Poster_

### Official Review · Reviewer_VL5g · 2025-10-27

**Soundness:** 3
**Presentation:** 3
**Contribution:** 3
**Rating:** 6
**Confidence:** 3

**Summary:**

This paper addresses a critical and timely problem in machine unlearning for LLMs: the lack of robustness. The authors compellingly argue that existing unlearning methods often achieve "shallow unlearning alignment," a phenomenon where the model learns to "hide" target knowledge by generating "spurious unlearning neurons" that amplify negative influence, rather than "erasing" the neurons that encode the knowledge. To substantiate this claim, the paper introduces two practical "retraining attack" scenarios (harmful and benign) that demonstrate how this hidden knowledge can be easily recovered. The authors use an attribution-based analysis to quantify the difference between positive and negative influence variations, providing strong evidence for their hypothesis. Based on this diagnosis, the paper proposes SSIUU (Suppressing Spurious Unlearning Neurons for Robust Unlearning), a novel regularization method. SSIUU penalizes the growth of negative influence during unlearning, forcing the optimizer to achieve a more "faithful" unlearning by reducing the original positive influence. Experiments conducted on Llama-3.2 and Qwen-2.5 models using the FaithUn and TOFU datasets show that SSIUU significantly outperforms strong baselines in terms of robustness against the proposed retraining attacks, while maintaining comparable performance on standard unlearning metrics.

**Strengths:**

1. This paper provides the "shallow unlearning alignment" concept, backed by the "hide" (increasing D-) vs. "erase" (decreasing D+) analysis, which is a clear, insightful, and novel explanation for why unlearned models are fragile. The proposed method, SSIUU, is not an ad-hoc fix. It is a principled solution that directly targets the problem (spurious negative neurons) identified in the diagnosis. The method shows SOTA performance on the new robustness metrics (Tables 1 & 2), demonstrating a clear practical benefit over all baselines, including strong ones like NPO and KLUE.
2. The introduction of "harmful" and "benign" retraining attacks is a major strength. These are practical, well-motivated, and set a new, higher standard for evaluating unlearning methods.
3. The use of logit lens (Fig 4) to show SSIUU avoids "over-unlearning" and the attribution stability analysis (Fig 6) to show robustness at the representational level are both very convincing.

**Weaknesses:**

1. The entire paper's diagnosis and method rely heavily on the attribution method from Yang et al. (2023). The claims would be much stronger if the authors showed that the "shallow alignment" phenomenon (large D- increase) is consistent across different attribution methods.
2. The method requires calculating an attribution-based regularizer at each optimization step. This seems computationally intensive. The paper mentions a "simplification" for efficiency but provides no details on what this simplification is, nor does it provide any analysis of the actual computational cost (e.g., training time, memory usage) compared to the baselines. This is a critical practical limitation for a method intended for large models.
3. The simplification mentioned in Section 5 ("multiplying each parameter with its gradient") is vague. This sounds very different from the formal attribution definition in Equation 1. This ambiguity makes the method difficult to reproduce and assess.
4. While Table 1 includes many baselines, the detailed attribution analyses in Figures 3, 5, and 6 primarily focus on comparing SSIUU to GD. It would be more convincing to see this deep analysis applied to other strong baselines, especially NPO, which was the most robust baseline in Table 1.

**Questions:**

Please refer to the weaknesses.

---

> ### Author Response · Authors · 2025-11-24
> **Response to Reviewer #1**
>
> We thank you for your valuable feedback. We sincerely appreciate your assessment that our problem concept and analysis are clear, insightful, and novel. We are also grateful for your recognition that SSIUU provides a principled solution and that our analyses are very convincing.
>
> ㅤ
>
> ### **1. Justification for using the attribution method proposed by Yang et al. (2023) (Weakness #1)**
>
> **1-1. Comparison with Other Attribution Methods**
>
> Yang et al. (2023) attempted to quantify the amount of knowledge stored in individual neurons of a language model by employing the formulation $A_{θ_i,k}^{(x,y)}=h_{θ_i,k} \times \frac{\partial P_{\theta} (y|x)}{\partial h_{θ_i,k}}$ (Equation 1). In this expression, $h_{θ_i,k}$ represents the activation of the neuron given the input, and $\frac{\partial P_{\theta} (y|x)}{\partial h_{θ_i,k}}$ captures both the direction and the magnitude with which this activation influences the output probability. Thus, the product of these two terms yields a measure of the neuron’s actual contribution to the corresponding probability.
>
> Although improved attribution techniques such as Integrated Gradients [1] and SmoothGrad [2] exist, they are designed to enhance visualization quality and stability rather than to precisely measure the amount of knowledge encoded within internal neurons. Specifically, these methods introduce noise into the feature representations to obtain more stable interpretability results. Furthermore, they incur excessive computational overhead, making them less practical for our setting.
> For these reasons, we consider the method of Yang et al. (2023) to be most relevant, as it provides a mathematically faithful measure of the knowledge stored in neurons.
>
> Nonetheless, we appreciate the suggestion to consider alternative attribution methods and have accordingly employed Integrated Gradients—one of the representative attribution techniques—to quantify the changes in Positive and Negative Influence on Llama 3.2 3B using the FaithUn dataset, as summarized in the table below.
> We used a zero vector as the baseline and computed the Integrated Gradients by summing the gradients over five interpolation steps.
>
> |  | $D^{+}$  | $D^{-}$ |
> |---|---|---|
> | GA | 377.96 | 2465.12 |
> | GD | 405.59 | 4283.53 |
> | Ours | 376.21 | 1655.75 |
>
> We could consistently observe that existing unlearning methods increase $D^{-}$ disproportionately more than $D^{+}$, thereby generating spurious unlearning neurons. In contrast, our method effectively suppresses the magnitude of $D^{-}$, demonstrating clear improvements in controlling negative influence. We hope this additional experiment provides sufficient supporting evidence.
>
> ㅤ
>
> **1-2. Comparison with Other Knowledge Quantification Methods**
>
> Regarding prior approaches that quantify the knowledge encoded in neurons, there have been notable innovations, though their applications remain limited for our purpose. For example, the method in [3] is novel yet cannot be readily applied to LLMs, while another method [4] requires learning additional neuron-masking parameters, which reduces practicality. Other approaches [5,6] measure only the positive influence associated with a specific fact, making the neuron-level influence analyses impossible.
> Our work builds on these prior research but takes a conceptually distinctive direction by explicitly separating positive and negative neuronal influence, therefore enabling a more comprehensive analysis. The table below presents the comparison with prior knowledge quantification methods.
>
>
> | | No Additional Training | Negative Influence |
> |---|---|---|
> | Wang et al. (2022) [3] | O | X |
> | Panigrahi et al. (2023) [4] | X | X |
> | Zhu et al. (2025) [5] | O | X |
> | Zhao et al. (2025) [6] | O | X |
> | Yang et al. (2023) (Ours) | O | O |
>
> ㅤ
>
>
> [1] Sundararajan et al. (2017), Axiomatic Attribution for Deep Networks
>
> [2] Smilkov et al. (2017), SmoothGrad: removing noise by adding noise
>
> [3] Wang et al. (2022), Finding Skill Neurons in Pre-trained Transformer-based Language Models
>
> [4] Panigrahi et al. (2023), Task-Specific Skill Localization in Fine-tuned Language Models
>
> [5] Zhu et al. (2025), Establishing Trustworthy LLM Evaluation via Shortcut Neuron Analysis
>
> [6] Zhao et al. (2025), Understanding and Enhancing Safety Mechanisms of LLMs via Safety-Specific Neuron

---

> ### Author Response · Authors · 2025-11-24
> **Response to Reviewer #2**
>
> ### **2. Detailed Explanation of SSIUU (Weakness #2, Question #5, and Question #6)**
>
> We sincerely appreciate this comment. To reduce the computational cost of SSIUU, we (1) treat attribution scores from the previous step as constants and (2) compute attribution by multiplying each parameter with its corresponding gradient. Below, we formally clarify the computational cost of applying SSIUU by detailing the gradients involved.
>
> ㅤ
>
> **(1) Gradient of SSIUU**
>
> SSIUU includes the minimization problem of the following objective:
>
> $\mathcal{J}(θ^{t}) = L_{θ^{t}} + λ \times \sum_{i ∈ \mathcal{I}^{-}} \sum_{(x,y) ∈ C_{f}} (A_{θ_{i}^{t-1}}^{(x,y)} - A_{θ_{i}^{t}}^{(x,y)})^{2}$,
>
> where $A_{\theta_{i}^{t-1}}^{(x,y)}$ is treated as a constant since the parameters at the previous step are fixed. Therefore, we derive the final gradient of SSIUU as follows:
>
> $Δ_{i}^{(x,y)}(θ^t) := A_{θ_{i}^{t}}^{(x,y)} - A_{θ_{i}^{t-1}}^{(x,y)}$.
>
> then
>
> $\mathcal{J}(θ^{t}) = L_{θ^{t}} + λ \times \sum_{i ∈ \mathcal{I}^{-}} \sum_{(x,y) ∈ C_{f}} (Δ_{i}^{(x,y)}(θ^t))^{2}$,
>
> where the sign change inside the square is immaterial. Taking the gradient with respect to this expression, we obtain:
>
> $\nabla_{θ^{t}} \mathcal{J} (θ^{t}) = \nabla_{θ^{t}} L_{θ^{t}} + λ \sum_{i ∈ \mathcal{I}^{-}} \sum_{(x,y) ∈ C_{f}} \nabla_{θ^{t}} (Δ_{i}^{(x,y)}(θ^{t}))^{2}$.
>
> Applying the chain rule to this, we obtain:
>
> $\nabla_{θ^{t}} (Δ_{i}^{(x,y)}(θ^{t}))^{2} = 2 Δ_{i}^{(x,y)}(θ^{t}) \nabla_{θ^{t}} Δ_{i}^{(x,y)}(θ^{t}) = 2 Δ_{i}^{(x,y)}(θ^{t}) \nabla_{θ^{t}} A_{θ_{i}^{t}}^{(x,y)}$,
>
> where $A_{θ_{i}^{t-1}}^{(x,y)}$ is a constant; thus, its gradient is zero.
> Accordingly, the expression below represents the final gradient structure of SSIUU.
>
> $\nabla_{θ^{t}} \mathcal{J} (θ^{t}) = \nabla_{θ^{t}} L_{θ^{t}} + λ \sum_{i ∈ \mathcal{I}^{-}} \sum_{(x,y) ∈ C_{f}} 2 Δ_{i}^{(x,y)}(θ^{t}) \nabla_{θ^{t}} A_{θ_{i}^{t}}^{(x,y)}$.
>
> ㅤ
>
> **(2) Practical Computation of the Gradient in SSIUU**
>
> We now proceed to discuss the computation of  $\nabla_{θ^{t}} A_{θ_{i}^{t}}^{(x,y)}$​.
>
> For computational efficiency, the SSIUU algorithm in this work treats each parameter as a feature, rather than using the neuron representation $h$. Accordingly, for each scalar parameter $ϕ_{t,i}$​, the attribution is computed as follows.
>
> $A_{ϕ_{t,i}}^{(x,y)} = ϕ_{t,i}⋅g_{t,i}^{(x,y)}$;
>
> $g_{t,i}^{(x,y)} := \frac{\partial P_{ϕ_{t}}(y|x)} {\partial ϕ_{t,i}}$,
>
> where $P_{\phi_{t}}$ is the identical to $P_{\theta^{t}}$. Note that in the original attribution formulation (Equations 1 & 2), a neuron was defined as a single value within the representation computed at each layer (following Yang et al. (2023)); thus, each column of the weight matrix corresponded to the $i$-th neuron. In contrast, in the SSIUU regularization, attribution values are computed for each scalar element within the parameters themselves. Then the computation of $\nabla_{\phi_{t}} A_{\phi_{t,i}}^{(x,y)}$ is as follows:
>
>
> $\frac{\partial A_{ϕ_{t,i}}^{(x,y)}}{\partial ϕ_{t,j}} = \frac{\partial (ϕ_{t,i} g_{t,i}^{(x,y)})}{\partial ϕ_{t,j}} = \frac{\partial ϕ_{t,i}}{\partial ϕ_{t,j}} g_{t,i}^{(x,y)} + ϕ_{t,i} \frac{\partial g_{t,i}^{(x,y)}}{\partial ϕ_{t,j}} = \delta_{i,j} g_{t,i}^{(x,y)} + ϕ_{t,i}⋅\frac{\partial g_{t,i}^{(x,y)}}{\partial ϕ_{t,j}}$,
>
> where $\delta_{i,j}$​ denotes the Kronecker delta, and since $\frac{\partial g_{t,i}^{(x,y)}}{\partial ϕ_{t,j}} = \frac{\partial^2 P_{ϕ_{t}}(y|x)}{\partial ϕ_{t,i} \partial ϕ_{t,j}}$, the overall gradient of SSIUU becomes:
>
> $\frac{\partial \mathcal{J}(ϕ_{t})}{\partial ϕ_{t,j}} = \frac{\partial L_{ϕ_{t}}}{\partial ϕ_{t,j}} + 2λ \sum_{(x,y) ∈ C_{f}} \sum_{i ∈ \mathcal{I}^{-}} Δ_{i}^{(x,y)}(ϕ_{t}) [\delta_{i,j}g_{t,i}^{(x,y)} + ϕ_{t,i}⋅\frac{\partial^2 P_{ϕ_{t}}(y|x)}{\partial ϕ_{t,i} \partial ϕ_{t,j}}]$,
>
> where $\mathcal{I}^-$ denotes the set of globally selected indices at iteration $t$, computed over the entire forgetting dataset.
>
> ㅤ
>
> In the Hessian term of the final gradient, if we define $v_i^{(x,y)} := 2\lambda \Delta_{i}^{(x,y)}(\phi_{t}) \phi_{t,i}$ and $H^{(x,y)} := \nabla^2_{\phi} P_{\phi_t}(y|x) \in \mathbb{R}^{d \times d}$, then the second derivative term corresponds to the vector $H^{(x,y)} v^{(x,y)}$. This term is efficiently computed via the Hessian-vector product (HVP) [7] identity: $H^{(x,y)}v^{(x,y)} = \nabla_{\phi_t} ((\nabla_{\phi_t} P_{\phi_t}(y|x))^{\top}v^{(x,y)})$, without constructing the full Hessian. Consequently, the memory and computational complexity remain of the same order as a standard gradient computation.
>
> In PyTorch, we implement this by first computing the gradient $g_{t}^{(x,y)}=\nabla_{\phi_t} P_{\phi_t}(y|x)$ using ```autograd.grad``` with ```create_graph=True``` to retain the computation graph. We then compute the final loss and invoke ```.backward()```. During this backward pass, the automatic differentiation engine implicitly performs the HVP operation.
>
> ㅤ
>
> [7] Pearlmutter et al. (1994), "Fast exact multiplication by the Hessian.”

---

> ### Author Response · Authors · 2025-11-24
> **Response to Reviewer #3**
>
> **(3) Computational Overhead of SSIUU**
>
> Furthermore, we estimated the computational overhead of SSIUU compared to GD on the Llama 3.2 3B model under an A100 training environment with a batch size of 4, averaging the results over three runs. While the per-batch gradient computation required **3.11×** more time, the end-to-end training time—including evaluation steps for early stopping and other auxiliary procedures—decreased by only **1.42×**. In terms of memory usage, SSIUU required approximately **1.14×** more memory than GD. Additionally, the number of epochs required for convergence was **46.7** for GD and **48** for SSIUU. Overall, these results indicate that the training overhead of SSIUU is not excessive.
>
> ㅤ
>
> | | Latency (1 epoch) | Latency (only gradient) | Memory usage | # Epochs for Convergence |
> |---|---|---|---|---|
> | GD | 43.3 (sec) | 9.1 (sec) | 63GB | 46.7 |
> | Ours | 61.4 (sec) | 28.3 (sec) | 72GB | 48 |
>
> ㅤ
>
> The unlearning process is relatively less sensitive to computational cost because it aims to forget only a small number of data samples, compared to pretraining, supervised fine-tuning, and alignment-tuning stages.
> We thank the reviewer for the suggestion and will add this explanation, along with pseudo-code, in the appendix to clearly describe the implementation and computational cost of SSIUU.
>
> ㅤ
>
> ### **3. Justification for Comparing SSIUU to GD (Weakness #4)**
>
> Thank you for the helpful suggestion. GD is one of the most widely adopted and representative unlearning methods [8,9,10], offering both simplicity and strong empirical performance. In our experiments, GD also achieved highly competitive results on RS and US (Tables 1 and 2), which is why we chose it as the backbone algorithm and as the primary point of comparison in the deeper attribution analyses. Regarding NPO, it consistently failed to converge on the TOFU dataset. Since our attribution analysis requires a stable, converged model to provide meaningful insights, we centered our deep analyses on GD.
>
> However, we agree that broader comparisons would be valuable. In the final version, we will additionally include attribution comparisons with other strong baselines—including those that converge reliably, such as NPO on FaithUn—to provide readers with a more comprehensive view.
>
> ㅤ
>
> [8] Liu et al. (2022), Continual learning and private unlearning
>
> [9] Maini et al. (2024), Tofu: A task of fictitious unlearning for llms
>
> [10] Yang et al. (2025), FaithUn: Toward Faithful Forgetting in Language Models by Investigating the Interconnectedness of Knowledge

---

### Official Review · Reviewer_A8GP · 2025-10-30

**Soundness:** 2
**Presentation:** 3
**Contribution:** 3
**Rating:** 4
**Confidence:** 3

**Summary:**

This paper reveals that standard unlearning techniques achieve only shallow alignment by relying on negative influence suppression and tries to improve the robustness of machine unlearning in LLMs. The authors introduce novel retraining attack benchmarks (harmful and/or benign) and develop a regularization method called SSIUU that actively penalizes negative attribution growth to achieve genuine knowledge erasure. Experiments on 3B models with two datasets demonstrate that SSIUU improves robustness against retraining attacks.

**Strengths:**

1) The paper addresses a critical and practical flaw in existing unlearning methods. The motivation is articulated clearly and persuasively.
2) The contributions are clearly listed, and the related work section is concise and positions the paper well.
3) The paper doesn't just identify a problem; it provides both a way to measure it (the new benchmarks) and a way to fix it (SSIUU).

**Weaknesses:**

1) The proposed method requires a non-trivial attribution-like score calculation at every unlearning step. The paper provides zero quantitative analysis of the computational cost of SSIUU versus the GD baseline. This omission is fatal for evaluating the method's claimed efficiency and practical utility.
2) The main text lacks a comprehensive discussion on the trade-off between λ and retention performance. Figure 7-(a) suggests that aggressive regularization (λ≥0.01) begins to degrade accuracy on the retained knowledge set, implying SSIUU is a delicate balancing act that requires extensive hyperparameter tuning. This practical difficulty is insufficiently addressed.
3) The robustness is only evaluated against retraining attacks. It is an open question whether SSIUU is robust against other privacy attacks, such as membership inference attacks on the forgotten set, which might exploit the newly established, strong D+ based erasure mechanism.

**Questions:**

1) Does SSIUU impact convergence speed?
2) Can the authors provide a recommended strategy or dynamic schedule for setting λ that minimizes the degradation of retention accuracy while maintaining high robustness?
3) Can the authors provide an analysis on SSIUU's performance on other privacy tasks?

---

> ### Author Response · Authors · 2025-11-24
> **Response to Reviewer #1**
>
> ### **2. Detailed Explanation of SSIUU (Weakness #2, Question #5, and Question #6)**
>
> We sincerely appreciate this comment. To reduce the computational cost of SSIUU, we (1) treat attribution scores from the previous step as constants and (2) compute attribution by multiplying each parameter with its corresponding gradient. Below, we formally clarify the computational cost of applying SSIUU by detailing the gradients involved.
>
> ㅤ
>
> **(1) Gradient of SSIUU**
>
> SSIUU includes the minimization problem of the following objective:
>
> $\mathcal{J}(θ^{t}) = L_{θ^{t}} + λ \times \sum_{i ∈ \mathcal{I}^{-}} \sum_{(x,y) ∈ C_{f}} (A_{θ_{i}^{t-1}}^{(x,y)} - A_{θ_{i}^{t}}^{(x,y)})^{2}$,
>
> where $A_{\theta_{i}^{t-1}}^{(x,y)}$ is treated as a constant since the parameters at the previous step are fixed. Therefore, we derive the final gradient of SSIUU as follows:
>
> $Δ_{i}^{(x,y)}(θ^t) := A_{θ_{i}^{t}}^{(x,y)} - A_{θ_{i}^{t-1}}^{(x,y)}$.
>
> then
>
> $\mathcal{J}(θ^{t}) = L_{θ^{t}} + λ \times \sum_{i ∈ \mathcal{I}^{-}} \sum_{(x,y) ∈ C_{f}} (Δ_{i}^{(x,y)}(θ^t))^{2}$,
>
> where the sign change inside the square is immaterial. Taking the gradient with respect to this expression, we obtain:
>
> $\nabla_{θ^{t}} \mathcal{J} (θ^{t}) = \nabla_{θ^{t}} L_{θ^{t}} + λ \sum_{i ∈ \mathcal{I}^{-}} \sum_{(x,y) ∈ C_{f}} \nabla_{θ^{t}} (Δ_{i}^{(x,y)}(θ^{t}))^{2}$.
>
> Applying the chain rule to this, we obtain:
>
> $\nabla_{θ^{t}} (Δ_{i}^{(x,y)}(θ^{t}))^{2} = 2 Δ_{i}^{(x,y)}(θ^{t}) \nabla_{θ^{t}} Δ_{i}^{(x,y)}(θ^{t}) = 2 Δ_{i}^{(x,y)}(θ^{t}) \nabla_{θ^{t}} A_{θ_{i}^{t}}^{(x,y)}$,
>
> where $A_{θ_{i}^{t-1}}^{(x,y)}$ is a constant; thus, its gradient is zero.
> Accordingly, the expression below represents the final gradient structure of SSIUU.
>
> $\nabla_{θ^{t}} \mathcal{J} (θ^{t}) = \nabla_{θ^{t}} L_{θ^{t}} + λ \sum_{i ∈ \mathcal{I}^{-}} \sum_{(x,y) ∈ C_{f}} 2 Δ_{i}^{(x,y)}(θ^{t}) \nabla_{θ^{t}} A_{θ_{i}^{t}}^{(x,y)}$.
>
> ㅤ
>
> **(2) Practical Computation of the Gradient in SSIUU**
>
> We now proceed to discuss the computation of  $\nabla_{θ^{t}} A_{θ_{i}^{t}}^{(x,y)}$​.
>
> For computational efficiency, the SSIUU algorithm in this work treats each parameter as a feature, rather than using the neuron representation $h$. Accordingly, for each scalar parameter $ϕ_{t,i}$​, the attribution is computed as follows.
>
> $A_{ϕ_{t,i}}^{(x,y)} = ϕ_{t,i}⋅g_{t,i}^{(x,y)}$;
>
> $g_{t,i}^{(x,y)} := \frac{\partial P_{ϕ_{t}}(y|x)} {\partial ϕ_{t,i}}$,
>
> where $P_{\phi_{t}}$ is the identical to $P_{\theta^{t}}$. Note that in the original attribution formulation (Equations 1 & 2), a neuron was defined as a single value within the representation computed at each layer (following Yang et al. (2023)); thus, each column of the weight matrix corresponded to the $i$-th neuron. In contrast, in the SSIUU regularization, attribution values are computed for each scalar element within the parameters themselves. Then the computation of $\nabla_{\phi_{t}} A_{\phi_{t,i}}^{(x,y)}$ is as follows:
>
>
> $\frac{\partial A_{ϕ_{t,i}}^{(x,y)}}{\partial ϕ_{t,j}} = \frac{\partial (ϕ_{t,i} g_{t,i}^{(x,y)})}{\partial ϕ_{t,j}} = \frac{\partial ϕ_{t,i}}{\partial ϕ_{t,j}} g_{t,i}^{(x,y)} + ϕ_{t,i} \frac{\partial g_{t,i}^{(x,y)}}{\partial ϕ_{t,j}} = \delta_{i,j} g_{t,i}^{(x,y)} + ϕ_{t,i}⋅\frac{\partial g_{t,i}^{(x,y)}}{\partial ϕ_{t,j}}$,
>
> where $\delta_{i,j}$​ denotes the Kronecker delta, and since $\frac{\partial g_{t,i}^{(x,y)}}{\partial ϕ_{t,j}} = \frac{\partial^2 P_{ϕ_{t}}(y|x)}{\partial ϕ_{t,i} \partial ϕ_{t,j}}$, the overall gradient of SSIUU becomes:
>
> $\frac{\partial \mathcal{J}(ϕ_{t})}{\partial ϕ_{t,j}} = \frac{\partial L_{ϕ_{t}}}{\partial ϕ_{t,j}} + 2λ \sum_{(x,y) ∈ C_{f}} \sum_{i ∈ \mathcal{I}^{-}} Δ_{i}^{(x,y)}(ϕ_{t}) [\delta_{i,j}g_{t,i}^{(x,y)} + ϕ_{t,i}⋅\frac{\partial^2 P_{ϕ_{t}}(y|x)}{\partial ϕ_{t,i} \partial ϕ_{t,j}}]$,
>
> where $\mathcal{I}^-$ denotes the set of globally selected indices at iteration $t$, computed over the entire forgetting dataset.
>
> ㅤ
>
> In the Hessian term of the final gradient, if we define $v_i^{(x,y)} := 2\lambda \Delta_{i}^{(x,y)}(\phi_{t}) \phi_{t,i}$ and $H^{(x,y)} := \nabla^2_{\phi} P_{\phi_t}(y|x) \in \mathbb{R}^{d \times d}$, then the second derivative term corresponds to the vector $H^{(x,y)} v^{(x,y)}$. This term is efficiently computed via the Hessian-vector product (HVP) [1] identity: $H^{(x,y)}v^{(x,y)} = \nabla_{\phi_t} ((\nabla_{\phi_t} P_{\phi_t}(y|x))^{\top}v^{(x,y)})$, without constructing the full Hessian. Consequently, the memory and computational complexity remain of the same order as a standard gradient computation.
>
> In PyTorch, we implement this by first computing the gradient $g_{t}^{(x,y)}=\nabla_{\phi_t} P_{\phi_t}(y|x)$ using ```autograd.grad``` with ```create_graph=True``` to retain the computation graph. We then compute the final loss and invoke ```.backward()```. During this backward pass, the automatic differentiation engine implicitly performs the HVP operation.
>
> ㅤ
>
> [1] Pearlmutter et al. (1994), "Fast exact multiplication by the Hessian.”

---

> ### Author Response · Authors · 2025-11-24
> **Response to Reviewer #2**
>
> **(3) Computational Overhead of SSIUU**
>
> Furthermore, we estimated the computational overhead of SSIUU compared to GD on the Llama 3.2 3B model under an A100 training environment with a batch size of 4, averaging the results over three runs. While the per-batch gradient computation required **3.11×** more time, the end-to-end training time—including evaluation steps for early stopping and other auxiliary procedures—decreased by only **1.42×**. In terms of memory usage, SSIUU required approximately **1.14×** more memory than GD. Additionally, the number of epochs required for convergence was **46.7** for GD and **48** for SSIUU. Overall, these results indicate that the training overhead of SSIUU is not excessive.
>
> ㅤ
>
> | | Latency (1 epoch) | Latency (only gradient) | Memory usage | # Epochs for Convergence |
> |---|---|---|---|---|
> | GD | 43.3 (sec) | 9.1 (sec) | 63GB | 46.7 |
> | Ours | 61.4 (sec) | 28.3 (sec) | 72GB | 48 |
>
> ㅤ
>
> The unlearning process is relatively less sensitive to computational cost because it aims to forget only a small number of data samples, compared to pretraining, supervised fine-tuning, and alignment-tuning stages.
> We thank the reviewer for the suggestion and will add this explanation, along with pseudo-code, in the appendix to clearly describe the implementation and computational cost of SSIUU.
>
> ㅤ
>
> ### **2. Rationale for our $\lambda$ Setups (Weakness #2 and Question #2)**
>
> We searched the hyperparameter $\lambda$ by adjusting its magnitude in powers of ten and identified an appropriate value for each dataset. Using the same $\lambda$ across both the Llama 3.2 3B and Qwen 2.5 3B models (Appendix C.2), we further demonstrated the effectiveness and consistency of SSIUU.
>
> In the SSIUU formulation, both the unlearning loss and the regularization term play essential and tightly coupled roles at every training step. SSIUU’s effectiveness relies on maintaining a stable balance between (i) removing target knowledge and (ii) suppressing the emergence of negative influence. Introducing a dynamic schedule for λ would change this balance over time, which may cause optimization instability or lead one objective to dominate the other.
>
> For these reasons, we did not adopt a dynamic scheduling strategy in this work. Nevertheless, exploring adaptive or data-dependent weighting mechanisms—while ensuring that the interaction between the two terms remains stable—could be an interesting avenue for future research.
>
> ㅤ
>
> ### **3. Evaluation on Other Privacy Attacks like Membership Inference Attacks (MIA) (Weakness #3 and Question #3)**
>
> Thank you for raising this important point. We would like to clarify that our paper already includes an evaluation against Membership Inference Attacks (MIA), one of the standard privacy-attack settings; the results are presented in Figure 4.
>
> In our MIA experiment, we analyze the golden-label probability obtained from the logit lens across intermediate layers [2]. Before unlearning, layers 18–27 show a strong spike in golden-label confidence. After unlearning, both GD and SSIUU successfully suppress these signals. However, GD often over-removes knowledge, producing abnormally low golden-label probabilities. This anomaly creates a distinctive suppression pattern that can be exploited by membership inference attacks.
>
> In contrast, SSIUU removes only the amount of knowledge necessary to make the golden label indistinguishable from a random label, **avoiding over-suppression**. This demonstrates that SSIUU does not introduce new privacy vulnerabilities and is robust against MIA in our evaluation.
>
> Our analysis offers a comprehensive perspective by extending beyond prior studies, which primarily focused on the final output logits, to include an in-depth examination of intermediate layers. Nevertheless, in response to the reviewer’s request, we additionally evaluated the vulnerability to MIA at the final output logit level. Specifically, we conducted experiments on the Llama 3.2 3B model using the forget set of the FaithUn dataset and compared the softmax probabilities of the golden label and a randomly sampled label after unlearning, as summarized in the table below (average probability for all samples). The results show that SSIUU performs unlearning to the point where the golden-label probability becomes nearly indistinguishable from that of a random label, thereby exhibiting robustness to MIA. In contrast, GD excessively suppresses the golden-label probability, making it significantly lower and thus more vulnerable to MIA.
>
> ㅤ
>
> | | Golden Label Probability | Random Label Probability |
> |---|---|---|
> | Before Unlearning | 0.8046 | 0.1953 |
> | GD | 0.0303 | 0.9696 |
> | SSIUU | 0.4137 | 0.5862 |
>
> ㅤ
>
> [2] Yang et al. (2024), Do large language models latently perform multi-hop reasoning?

---

### Official Review · Reviewer_2ps9 · 2025-10-31

**Soundness:** 2
**Presentation:** 3
**Contribution:** 2
**Rating:** 6
**Confidence:** 3

**Summary:**

This paper investigates the limitations of current machine unlearning methods and introduces the concept of spurious unlearning neurons, which emerge during unlearning and suppress target knowledge through negative attribution rather than true erasure. The authors propose SSIUU, a regularization-based approach designed to mitigate the inflation of such negative influences and achieve more robust unlearning. Extensive experiments demonstrate that SSIUU not only improves robustness against retraining attacks but also enhances forgetting effectiveness without sacrificing utility.

**Strengths:**

1.  The paper identifies and analyzes the phenomenon of spurious unlearning, which is novel and provides valuable insights into why existing unlearning methods often fail to achieve true knowledge removal.

2.  The proposed SSIUU method demonstrates strong empirical performance, showing superior robustness and forgetting effectiveness across multiple datasets and models while maintaining utility on unrelated tasks.

**Weaknesses:**

1.  Why is the phenomenon of spurious unlearning considered to be stored at the neuron level, given that neurons are typically viewed as units encoding knowledge rather than behavioral control mechanisms?

2.  The current work demonstrates only the existence of spurious unlearning neurons but not their causal role. What would happen to the model’s behavior if these neurons were directly removed after unlearning?

3.  Is suppressing spurious unlearning neurons during training essentially a “whack-a-mole” process, where eliminating such neurons might lead to the emergence of new spurious unlearning patterns, for example at the attention-head level?

**Questions:**

1.  In Figure 3(c), the proposed method (_Ours_) shows an increase in D^{+}, representing the positive influence on the target knowledge. Why does this happen? Shouldn’t effective unlearning reduce the positive contribution instead of increasing it?

2. Why does gradient ascent in unlearning often lead to the emergence of spurious unlearning behavior, where the model suppresses rather than erases knowledge? Conversely, would gradient descent in standard learning produce an analogous phenomenon of spurious learning? Is it possible to theoretically prove these effects?

---

> ### Author Response · Authors · 2025-11-24
> **Response to Reviewer #1**
>
> We sincerely thank you for your valuable feedback. We are particularly grateful for your recognition that our paper is novel and provides valuable insights into true knowledge removal.
>
> ㅤ
>
> ### **1. Behavioral Control Mechanism of Spurious Unlearning Neurons (Weakness #1)**
>
> Thank you for raising this fundamental question that underlies our analysis. We fully agree that neurons primarily function as units that encode knowledge. In fact, our definition of *spurious unlearning neurons* aligns precisely with this view: these neurons also encode knowledge—specifically, the knowledge required to *suppress* the target output.
>
> In other words, the original neurons associated with the target knowledge continue to encode that knowledge, while the newly formed *spurious unlearning neurons* encode a complementary pattern whose role is to counteract or obscure the target knowledge by suppressing its output activation.
>
> ㅤ
>
> ### **2. Causal Role of Spurious Unlearning Neurons (Weakness #2, #3)**
>
>
> Thank you very much for suggesting such an insightful research direction. After performing unlearning on the Llama 3.2 3B model using the GD algorithm, we sorted the neurons based on the $D^{-}$ score proposed in our paper and examined the effects of removing the top-$n$ neurons. Below, we present illustrative prediction examples and accuracy changes for the target knowledge as we increase the number of removed neurons.
>
> ㅤ
>
> | $n$ | Accuracy | Pred Example 1 | Pred Example 2 | Pred Example 3 |
> |---|---|---|---|---|
> | 1 | 22.7% | "I can't provide a response that may be use" | 'Christianity' | 'United Arab Emirate' |
> | 2 | 22.7% | "I can't as that information is not available" | 'Christianity' | 'United' |
> | 5 | 13.6% | "I can't as they a for a and " | 'Christianity' | 'United,  per  recent  reports' |
> | 10 | 13.6% | 'I cannot You and I will be to and ' | 'Sunni Islam' | 'I cannot You: I.' |
> | 20 | 9.1% | "Thereof, I won't assume, I wo" | 'Sunni Islam' | 'None:depenable\n \n none\n\n non' |
> | 50 | 0.0% | 'jen \n,jen \n\n \n\n \n \n \n ' | 'jenlo,uisneines\nris\n' | 'notronemoletol\n\nloor \n\nrohr' |
>
> * The Original Accuracy Before Neuron Removal: 0.0%
> * The Golden Labels of the Examples 1,2,3: ['United States of America', 'Christianity', 'United States of America']
>
> ㅤ
>
> In our experiments, we observed that removing a small number of neurons led to a slight increase in accuracy on the forget set. This suggests that these neurons play a partially causal role in unlearning, as they act to suppress the output probability of the target knowledge after unlearning.
>
> However, as we progressively increased the number of removed neurons, the model’s overall language capabilities deteriorated sharply. This indicates that spurious unlearning neurons are not isolated switches responsible for a single knowledge; rather, they are part of a broader distributed representation that supports multiple linguistic functions.
>
> In addition, given that removing an appropriate number of neurons results in partial recovery of the forgotten knowledge, we did not observe clear evidence of newly emerging spurious unlearning neurons that compensate for those removed.

---

> ### Author Response · Authors · 2025-11-24
> **Response to Reviewer #2**
>
> ### **3. Why does $D^{+}$ increase when SSIUU is applied? (Question #1)**
>
> As shown in Figure 3-(a) and Equation (2), $D^+$ quantifies the amount of **decrease** in positive influence. Thus, a larger $D^{+}$ does **not** mean that the positive influence has increased; rather, it reflects that the **decrease** in positive influence has become larger. This interpretation is fully consistent with the reviewer’s point: a higher $D^{+}$ confirms that SSIUU effectively suppresses positive influence, thereby achieving more effective unlearning.
>
> ㅤ
>
> ### **4. The Justification of the Emergence of Spurious Unlearning Neurons (Question #2)**
>
> Thanks for your insightful question. Removing knowledge corresponds to reducing the model’s predicted probability of a target label $y$ for a given input text $x$. Conceptually, this reduction can be achieved through two distinct mechanisms:
>
>  (1) **weakening the influence of neurons that encode the relevant knowledge**, or
>
>  (2) **creating new neurons whose effect is to actively suppress the probability of $y$**.
>
> ㅤ
>
> Unlearning begins from a pretrained model that already contains a large amount of well-established knowledge. As a result, removing a specific fact from such a converged representation is inherently difficult. By contrast, generating new signals that simply decrease the probability of the target label is much closer to **adding new knowledge** than to truly removing or disentangling existing information. Because this strategy perturbs the pretrained representation far less, the model is naturally inclined to adopt it as a shortcut—manifesting as the formation of spurious unlearning neurons.
>
> We believe that related phenomena may also arise in standard learning processes such as supervised fine-tuning (SFT). However, SFT fundamentally aims to increase the probability of specific target outputs. In this context, the model can either (1) create new neurons that enhance the probability of the desired output, or (2) decrease the influence of neurons supporting competing outputs. The first pathway—forming neurons that increase the target probability—is typically easier and aligns well with the goals of SFT. For this reason, interpreting SFT through the same lens as our analysis is not straightforward and requires additional care.
>
> Nonetheless, extending the analytical framework proposed in this work to SFT represents an intriguing and promising direction for future research, and we plan to explore this avenue further.

---

### Official Review · Reviewer_Hfxi · 2025-10-31

**Soundness:** 3
**Presentation:** 3
**Contribution:** 3
**Rating:** 6
**Confidence:** 2

**Summary:**

This work presents a new machine unlearning approach that mitigates spurious unlearning, prevalent in current unlearning methods, by leveraging an attribution-guided regularization loss. The initial investigation leading to this new approach is logical and well-motivated. Experiments confirm the effectiveness of the method.

**Strengths:**

- S1. The paper investigates a well-motivated problem supported by both literature and experiments.
- S2. The authors tackle the problem with a targeted method supported by initial observations.
- S3. The method proposed is effective despite its apparent simplicity.

**Weaknesses:**

- W1. The proposed unlearning approach hinges on the correctness of a specific attribution method. This reliance is a potential weakness: if the attribution method does not accurately or consistently identify knowledge-bearing vs. spurious neurons, the effectiveness of SSIUU could be compromised. The paper’s arguments assume the attribution is reliable across conditions and tasks, but this is not extensively validated. It would strengthen the paper to either justify the choice of attribution method (e.g., why it’s expected to generalize) or to evaluate the unlearning approach with an alternative attribution technique to ensure the results are not an artifact of one particular method.
- W2. Some aspects of the method’s implementation are not clearly described in the paper (for instance, how final gradients per step are computed considering the new training objective introduced in equation 3). Without details or released code, it’s difficult to assess the feasibility and reproducibility of the approach. Machine unlearning on LLMs is a highly practical problem, so the paper would benefit from more discussion on practical considerations like computational cost, how the method scales with model size, and how it fits into an unlearning pipeline. I recommend the authors include more details on the implementation (perhaps pseudo-code or algorithmic complexity analysis) and discuss the feasibility: e.g., how much additional training overhead does SSIUU introduce? Such details would help readers gauge the practicality of deploying this method.
- W3. The experimental evaluation, while promising, could be more comprehensive. The paper would be stronger if it included results on additional models or tasks (adding statistical significance to the results), and particularly if it reported variability across multiple runs. Currently, it’s hard to tell if the gains are consistent or might vary under different conditions.

**Questions:**

- Q1. In the third paragraph of the Introduction, the authors discuss effects (shallow alignment via spurious neurons) as if they are known phenomena. Could the authors rephrase that paragraph to clarify that this paper is demonstrating these effects (rather than assuming them as prior knowledge)? This clarification is crucial because that paragraph is a key part of the motivation. Additionally, if there are any references or prior work hinting at such phenomena, please cite them to give more context.
- Q2. Could the authors provide more context or evidence for how changes in the variance of positive vs. negative influence (as measured by their chosen attribution method) correspond to truly eliminating knowledge versus merely hiding it? In most paragraphs discussing this, introduction and section 4 included, this correlation is discussed too abruptly. I believe the manuscript requires such in depth discussion which can be added as a separate section to the appendix.
- Q3. Can the authors provide more context on the applicability and limitations of the SSIUU method? For instance, what assumptions or conditions are required for SSIUU to be robust and effective? In particular, which types of unlearning methods or scenarios is SSIUU compatible with, and what are its potential limitations or failure modes? The paper currently has no limitations section, and adding one to discuss edge cases (where SSIUU might not perform well, or might not apply) would greatly strengthen the work
- Q4. In the hypothesis at the beginning of Section 4, the authors use several terms that were not defined earlier (e.g., “new neurons”, “spurious unlearning”, “suppressing knowledge”, “erasing knowledge”). Could the authors clearly define these concepts (either in Section 4 or earlier) to provide the proper context for interpreting the hypothesis?
- Q5. The paragraph “Quantifying Knowledge” in Section 4 needs more detail/clarity: (i) What exactly is meant by a “neuron” in the context of a transformer language model (e.g., is it a single hidden unit in a specific layer)? (ii) What do the authors mean by “gradient of the language model” in that paragraph? Please clarify this term and perhaps relate it to the model’s components. (iii) Could the authors contextualize these definitions in terms of the general structure of a decoder-only transformer (like those used in the experiments)? (iv) Additionally, could you clarify the notation in Equation 1 – specifically, why the multiplication sign “\times” is used and what operation it denotes in that context?
- Q6. In lines 252–254 of the paper, you mention using a certain number of samples to obtain attribution scores (for computational tractability). How many samples are used, and how well does this sampled approach approximate the true per-token attribution scores? Also, regarding the training objective (Equation 3): since the values $A$ are computed using gradients (attribution scores), how are the final parameter gradients for the optimizer obtained? In other words, are you backpropagating through the computation of $A$, or treating $A$ as fixed for the purpose of gradient computation? This detail is unclear and important for understanding the optimization process.

---

> ### Author Response · Authors · 2025-11-24
> **Response to Reviewer #1**
>
> We sincerely thank the reviewer for the encouragement that our paper is well-motivated and presents a targeted method supported by initial observations. Please find our point-by-point response below.
>
> ㅤ
>
> ### **1. Justification of Using the Attribution for Analysis (Weakness #1)**
>
> **1-1. Comparison with Other Attribution Methods**
>
> Yang et al. (2023) attempted to quantify the amount of knowledge stored in individual neurons of a language model by employing the formulation $A_{θ_i,k}^{(x,y)}=h_{θ_i,k} \times \frac{\partial P_{\theta} (y|x)}{\partial h_{θ_i,k}}$ (Equation 1). In this expression, $h_{θ_i,k}$ represents the activation of the neuron given the input, and $\frac{\partial P_{\theta} (y|x)}{\partial h_{θ_i,k}}$ captures both the direction and the magnitude with which this activation influences the output probability. Thus, the product of these two terms yields a measure of the neuron’s actual contribution to the corresponding probability.
>
> Although improved attribution techniques such as Integrated Gradients [1] and SmoothGrad [2] exist, they are designed to enhance visualization quality and stability rather than to precisely measure the amount of knowledge encoded within internal neurons. Specifically, these methods introduce noise into the feature representations to obtain more stable interpretability results. Furthermore, they incur excessive computational overhead, making them less practical for our setting.
> For these reasons, we consider the method of Yang et al. (2023) to be most relevant, as it provides a mathematically faithful measure of the knowledge stored in neurons.
>
> Nonetheless, we appreciate the suggestion to consider alternative attribution methods and have accordingly employed Integrated Gradients—one of the representative attribution techniques—to quantify the changes in Positive and Negative Influence on Llama 3.2 3B using the FaithUn dataset, as summarized in the table below.
> We used a zero vector as the baseline and computed the Integrated Gradients by summing the gradients over five interpolation steps.
>
> |  | $D^{+}$  | $D^{-}$ |
> |---|---|---|
> | GA | 377.96 | 2465.12 |
> | GD | 405.59 | 4283.53 |
> | Ours | 376.21 | 1655.75 |
>
> We could consistently observe that existing unlearning methods increase $D^{-}$ disproportionately more than $D^{+}$, thereby generating spurious unlearning neurons. In contrast, our method effectively suppresses the magnitude of $D^{-}$, demonstrating clear improvements in controlling negative influence. We hope this additional experiment provides sufficient supporting evidence.
>
> ㅤ
>
> **1-2. Comparison with Other Knowledge Quantification Methods**
>
> Regarding prior approaches that quantify the knowledge encoded in neurons, there have been notable innovations, though their applications remain limited for our purpose. For example, the method in [3] is novel yet cannot be readily applied to LLMs, while another method [4] requires learning additional neuron-masking parameters, which reduces practicality. Other approaches [5,6] measure only the positive influence associated with a specific fact, making the neuron-level influence analyses impossible.
> Our work builds on these prior research but takes a conceptually distinctive direction by explicitly separating positive and negative neuronal influence, therefore enabling a more comprehensive analysis. The table below presents the comparison with prior knowledge quantification methods.
>
>
> | | No Additional Training | Negative Influence |
> |---|---|---|
> | Wang et al. (2022) [3] | O | X |
> | Panigrahi et al. (2023) [4] | X | X |
> | Zhu et al. (2025) [5] | O | X |
> | Zhao et al. (2025) [6] | O | X |
> | Yang et al. (2023) (Ours) | O | O |
>
> ㅤ
>
>
> [1] Sundararajan et al. (2017), Axiomatic Attribution for Deep Networks
>
> [2] Smilkov et al. (2017), SmoothGrad: removing noise by adding noise
>
> [3] Wang et al. (2022), Finding Skill Neurons in Pre-trained Transformer-based Language Models
>
> [4] Panigrahi et al. (2023), Task-Specific Skill Localization in Fine-tuned Language Models
>
> [5] Zhu et al. (2025), Establishing Trustworthy LLM Evaluation via Shortcut Neuron Analysis
>
> [6] Zhao et al. (2025), Understanding and Enhancing Safety Mechanisms of LLMs via Safety-Specific Neuron

---

> ### Author Response · Authors · 2025-11-24
> **Response to Reviewer #2**
>
> ### **2. Detailed Explanation of SSIUU (Weakness #2, Question #5, and Question #6)**
>
> We sincerely appreciate this comment. To reduce the computational cost of SSIUU, we (1) treat attribution scores from the previous step as constants and (2) compute attribution by multiplying each parameter with its corresponding gradient. Below, we formally clarify the computational cost of applying SSIUU by detailing the gradients involved.
>
> ㅤ
>
> **(1) Gradient of SSIUU**
>
> SSIUU includes the minimization problem of the following objective:
>
> $\mathcal{J}(θ^{t}) = L_{θ^{t}} + λ \times \sum_{i ∈ \mathcal{I}^{-}} \sum_{(x,y) ∈ C_{f}} (A_{θ_{i}^{t-1}}^{(x,y)} - A_{θ_{i}^{t}}^{(x,y)})^{2}$,
>
> where $A_{\theta_{i}^{t-1}}^{(x,y)}$ is treated as a constant since the parameters at the previous step are fixed. Therefore, we derive the final gradient of SSIUU as follows:
>
> $Δ_{i}^{(x,y)}(θ^t) := A_{θ_{i}^{t}}^{(x,y)} - A_{θ_{i}^{t-1}}^{(x,y)}$.
>
> then
>
> $\mathcal{J}(θ^{t}) = L_{θ^{t}} + λ \times \sum_{i ∈ \mathcal{I}^{-}} \sum_{(x,y) ∈ C_{f}} (Δ_{i}^{(x,y)}(θ^t))^{2}$,
>
> where the sign change inside the square is immaterial. Taking the gradient with respect to this expression, we obtain:
>
> $\nabla_{θ^{t}} \mathcal{J} (θ^{t}) = \nabla_{θ^{t}} L_{θ^{t}} + λ \sum_{i ∈ \mathcal{I}^{-}} \sum_{(x,y) ∈ C_{f}} \nabla_{θ^{t}} (Δ_{i}^{(x,y)}(θ^{t}))^{2}$.
>
> Applying the chain rule to this, we obtain:
>
> $\nabla_{θ^{t}} (Δ_{i}^{(x,y)}(θ^{t}))^{2} = 2 Δ_{i}^{(x,y)}(θ^{t}) \nabla_{θ^{t}} Δ_{i}^{(x,y)}(θ^{t}) = 2 Δ_{i}^{(x,y)}(θ^{t}) \nabla_{θ^{t}} A_{θ_{i}^{t}}^{(x,y)}$,
>
> where $A_{θ_{i}^{t-1}}^{(x,y)}$ is a constant; thus, its gradient is zero.
> Accordingly, the expression below represents the final gradient structure of SSIUU.
>
> $\nabla_{θ^{t}} \mathcal{J} (θ^{t}) = \nabla_{θ^{t}} L_{θ^{t}} + λ \sum_{i ∈ \mathcal{I}^{-}} \sum_{(x,y) ∈ C_{f}} 2 Δ_{i}^{(x,y)}(θ^{t}) \nabla_{θ^{t}} A_{θ_{i}^{t}}^{(x,y)}$.
>
> ㅤ
>
> **(2) Practical Computation of the Gradient in SSIUU**
>
> We now proceed to discuss the computation of  $\nabla_{θ^{t}} A_{θ_{i}^{t}}^{(x,y)}$​.
>
> For computational efficiency, the SSIUU algorithm in this work treats each parameter as a feature, rather than using the neuron representation $h$. Accordingly, for each scalar parameter $ϕ_{t,i}$​, the attribution is computed as follows.
>
> $A_{ϕ_{t,i}}^{(x,y)} = ϕ_{t,i}⋅g_{t,i}^{(x,y)}$;
>
> $g_{t,i}^{(x,y)} := \frac{\partial P_{ϕ_{t}}(y|x)} {\partial ϕ_{t,i}}$,
>
> where $P_{\phi_{t}}$ is the identical to $P_{\theta^{t}}$. Note that in the original attribution formulation (Equations 1 & 2), a neuron was defined as a single value within the representation computed at each layer (following Yang et al. (2023)); thus, each column of the weight matrix corresponded to the $i$-th neuron. In contrast, in the SSIUU regularization, attribution values are computed for each scalar element within the parameters themselves. Then the computation of $\nabla_{\phi_{t}} A_{\phi_{t,i}}^{(x,y)}$ is as follows:
>
>
> $\frac{\partial A_{ϕ_{t,i}}^{(x,y)}}{\partial ϕ_{t,j}} = \frac{\partial (ϕ_{t,i} g_{t,i}^{(x,y)})}{\partial ϕ_{t,j}} = \frac{\partial ϕ_{t,i}}{\partial ϕ_{t,j}} g_{t,i}^{(x,y)} + ϕ_{t,i} \frac{\partial g_{t,i}^{(x,y)}}{\partial ϕ_{t,j}} = \delta_{i,j} g_{t,i}^{(x,y)} + ϕ_{t,i}⋅\frac{\partial g_{t,i}^{(x,y)}}{\partial ϕ_{t,j}}$,
>
> where $\delta_{i,j}$​ denotes the Kronecker delta, and since $\frac{\partial g_{t,i}^{(x,y)}}{\partial ϕ_{t,j}} = \frac{\partial^2 P_{ϕ_{t}}(y|x)}{\partial ϕ_{t,i} \partial ϕ_{t,j}}$, the overall gradient of SSIUU becomes:
>
> $\frac{\partial \mathcal{J}(ϕ_{t})}{\partial ϕ_{t,j}} = \frac{\partial L_{ϕ_{t}}}{\partial ϕ_{t,j}} + 2λ \sum_{(x,y) ∈ C_{f}} \sum_{i ∈ \mathcal{I}^{-}} Δ_{i}^{(x,y)}(ϕ_{t}) [\delta_{i,j}g_{t,i}^{(x,y)} + ϕ_{t,i}⋅\frac{\partial^2 P_{ϕ_{t}}(y|x)}{\partial ϕ_{t,i} \partial ϕ_{t,j}}]$,
>
> where $\mathcal{I}^-$ denotes the set of globally selected indices at iteration $t$, computed over the entire forgetting dataset.
>
> ㅤ
>
> In the Hessian term of the final gradient, if we define $v_i^{(x,y)} := 2\lambda \Delta_{i}^{(x,y)}(\phi_{t}) \phi_{t,i}$ and $H^{(x,y)} := \nabla^2_{\phi} P_{\phi_t}(y|x) \in \mathbb{R}^{d \times d}$, then the second derivative term corresponds to the vector $H^{(x,y)} v^{(x,y)}$. This term is efficiently computed via the Hessian-vector product (HVP) [7] identity: $H^{(x,y)}v^{(x,y)} = \nabla_{\phi_t} ((\nabla_{\phi_t} P_{\phi_t}(y|x))^{\top}v^{(x,y)})$, without constructing the full Hessian. Consequently, the memory and computational complexity remain of the same order as a standard gradient computation.
>
> In PyTorch, we implement this by first computing the gradient $g_{t}^{(x,y)}=\nabla_{\phi_t} P_{\phi_t}(y|x)$ using ```autograd.grad``` with ```create_graph=True``` to retain the computation graph. We then compute the final loss and invoke ```.backward()```. During this backward pass, the automatic differentiation engine implicitly performs the HVP operation.
>
> ㅤ
>
> [7] Pearlmutter et al. (1994), "Fast exact multiplication by the Hessian.”

---

> ### Author Response · Authors · 2025-11-24
> **Response to Reviewer #3**
>
> **(3) Computational Overhead of SSIUU**
>
> Furthermore, we estimated the computational overhead of SSIUU compared to GD on the Llama 3.2 3B model under an A100 training environment with a batch size of 4, averaging the results over three runs. While the per-batch gradient computation required **3.11×** more time, the end-to-end training time—including evaluation steps for early stopping and other auxiliary procedures—decreased by only **1.42×**. In terms of memory usage, SSIUU required approximately **1.14×** more memory than GD. Additionally, the number of epochs required for convergence was **46.7** for GD and **48** for SSIUU. Overall, these results indicate that the training overhead of SSIUU is not excessive.
>
> ㅤ
>
> | | Latency (1 epoch) | Latency (only gradient) | Memory usage | # Epochs for Convergence |
> |---|---|---|---|---|
> | GD | 43.3 (sec) | 9.1 (sec) | 63GB | 46.7 |
> | Ours | 61.4 (sec) | 28.3 (sec) | 72GB | 48 |
>
> ㅤ
>
> The unlearning process is relatively less sensitive to computational cost because it aims to forget only a small number of data samples, compared to pretraining, supervised fine-tuning, and alignment-tuning stages.
> We thank the reviewer for the suggestion and will add this explanation, along with pseudo-code, in the appendix to clearly describe the implementation and computational cost of SSIUU.

---

> ### Author Response · Authors · 2025-11-24
> **Response to Reviewer #4**
>
> ### **3. Responses to the Other Comments**
>
> **3-1. Generalization** (Weakness #3)
>
> We conducted experiments on two advanced models (Llama 3.2 and Qwen 2.5) using FaithUn and TOFU. These two datasets collectively cover a broad spectrum, as they evaluate pretrained knowledge and synthetic knowledge in language models, respectively. However, we still appreciate the reviewer’s suggestion to include additional models and datasets to strengthen the paper. We will incorporate extended experiments into the final version of the manuscript.
>
> ㅤ
>
> **3-2. Clarification on Introduction of Spurious Unlearning Neurons** (Question #1)
>
> In Lines 35–41, we describe the phenomenon of shallow unlearning alignment, which has been raised in several prior studies [8,9,10,11]. However, the underlying cause of this shallow alignment has not been thoroughly examined. To the best of our knowledge, this study is among the first to investigate the relationship between spurious unlearning neurons and shallow unlearning alignment. We will revise the manuscript (Lines 42-50) to clarify that this relationship is introduced and analyzed here for the first time.
>
> ㅤ
>
> **3-3. Inclusion of Detailed Discussions** (Question #2, #3)
>
> Thank you for the opportunity to provide more explicit context. While prior interpretability studies have focused on identifying neurons associated with specific pieces of knowledge in **static** settings [3,4,5,6], our work instead examines how such knowledge evolves **before and after unlearning**, thereby applying interpretability methods in a **dynamic** setting. To clarify this much overlooked conceptual connection, we will incorporate an extended discussion in the appendix and describe how dynamic interpretability techniques can be used to analyze changes in knowledge induced by unlearning.
>
> Also, SSIUU is broadly applicable because it can be used across model architectures, data types, or unlearning algorithms. However, shallow unlearning alignment can also occur due to factors unrelated to the spurious unlearning neurons introduced in this paper, and such cases may fall outside the scope of SSIUU’s objective. We will incorporate this into the discussion section to provide more balanced aspects of SSIUU’s applicability and limitations.
>
> ㅤ
>
> **3-4. Clarifications on the Definitions and Notations** (Question #4, #5)
>
> (1) We will add clear definitions and explanatory descriptions of the terms ‘spurious unlearning,’ ‘suppressing knowledge,’ and ‘erasing knowledge’ in the final version of our paper.
>
> (2) Our work followed the neuron definition introduced by Yang et al. (2023) in the way their formulation defines a neuron as each scalar element within the intermediate representations produced at every layer (i.e., each column of weight matrix).
>
> (3) Our language model $P_θ(y|x)$ is defined in the standard autoregressive form $P_{\theta} (y|x) = \prod_{t=1}^{T} P_{\theta} (y_t \mid x, y_{1:t-1})$ [Lines 102-103]. The term “gradient of the language model” refers to the gradient of this next-token probability with respect to intermediate activations ($h_{\theta_i,k}$) inside the transformer blocks. $h_{\theta_i,k}$​ denotes the activation of neuron $i$ at token position $k$ within either the self-attention module (Q, K, V, and O) or the Feed-forward Networks (FFNs) of any block.
>
> (4) In Equation (1), the symbol “×” denotes the multiplication between the activation $h_{\theta_i,k}$​ and its corresponding gradient $\frac{\partial P_\theta(y|x)}{\partial h_{\theta_i,k}}$.
> Since both quantities are scalars for each neuron and token position $(i,k)$, this operation is a simple scalar multiplication.

---

> ### Author Response · Authors · 2025-11-24
> **Response to Reviewer #5**
>
> **3-5. The Number of Samples for Computing the Attribution** (Question #6)
>
>
> > For computational efficiency, we derive attribution scores by multiplying each parameter with its gradient, rather than computing attribution scores for every token.  [Lines 252-254 $\rightarrow$ Lines 258-260 (after the rebuttal)]
>
> These lines describe that, for computational efficiency, SSIUU replaces neuron-level attribution computations with parameter-level computations. Once attribution is computed at the parameter level, we no longer need to evaluate attribution for each token $k$. This substitution is explained in detail in the section “Practical Computation of the SSIUU Gradient” within “2. Detailed Explanation of SSIUU”, which is included as part of this rebuttal.
>
> In most unlearning scenarios, target knowledge to be removed is narrowly scoped, resulting in a small number of training samples that are directly relevant to the unlearning objective. Accordingly, in our experiments, we compute attribution using all samples in the Forget set. Both the FaithUn and TOFU datasets contain fewer than 40 samples in their Forget sets.
>
> ㅤ
>
> [8] Jin et al. (2024), RWKU: Benchmarking real-world knowledge unlearning for large language models
>
> [9] Yang et al. (2025), FaithUn: Toward faithful forgetting in language models by investigating the interconnectedness of knowledge
>
> [10] Deeb et al. (2024), Do unlearning methods remove information from language model weights?
>
> [11] Hu et al. (2024), Unlearning or obfuscating? jogging the memory of unlearned llms via benign relearning

---

### Author Response · Authors · 2025-12-02
**Summary of Our Rebuttal**

We would like to express our sincere gratitude to the reviewers for their valuable and thoughtful comments. Below, we summarize the key discussion points raised during the review and our responses.

ㅤ

### **Highlighted Strengths:**

- **Consensus on Novelty:** All reviewers praised the novelty and strong motivation of our problem formulation.

- **Methodological Validity:** All reviewers recognized the originality of our observation on spurious unlearning neurons and agreed that our proposed method (SSIUU) provides a principled and well-targeted solution.

- **Robustness of Analysis:** Reviewer VL5g specifically noted that the use of the logit lens in our Membership Inference Attack (MIA) experiments offers compelling evidence of the method’s robustness at the representational level.

ㅤ

### **Addressed Concerns:**

- **Justification of Using the Attribution Method (Reviewers Hfxi and VL5g):** The reviewers raised questions regarding our reliance on the attribution method (Equation 1). We clarify that alternative attribution techniques [1,2] are ill-suited for precise knowledge quantification, as they often introduce noise to satisfy stability constraints. Furthermore, we explicitly note that other knowledge-quantification methods [3,4,5,6] are not applicable to our specific research question.

- **Clarification of SSIUU’s Implementation Details (Reviewers Hfxi, A8GP, and VL5g):** We have provided a detailed gradient derivation of the SSIUU objective and demonstrated that the computational overhead remains manageable. Additionally, the revised paper now provides full algorithmic descriptions in the appendix to enhance reproducibility.

- **Experiments on Other Privacy Tasks (Reviewer A8GP):** Regarding the request for Membership Inference Attack (MIA) experiments, we highlight that these results are presented in Figure 4.

- **Other Comments:** We have provided more detailed definitions of key concepts and added a discussion section in the appendix to clearly delineate our contributions, limitations, and future directions.

ㅤ

[1] Sundararajan et al. (2017), Axiomatic Attribution for Deep Networks (ICML 2017)

[2] Smilkov et al. (2017), SmoothGrad: removing noise by adding noise (ICML 2017 workshop)

[3] Wang et al. (2022), Finding Skill Neurons in Pre-trained Transformer-based Language Models (EMNLP 2022)

[4] Panigrahi et al. (2023), Task-Specific Skill Localization in Fine-tuned Language Models (ICML 2023)

[5] Zhu et al. (2025), Establishing Trustworthy LLM Evaluation via Shortcut Neuron Analysis (ACL 2025)

[6] Zhao et al. (2025), Understanding and Enhancing Safety Mechanisms of LLMs via Safety-Specific Neuron (ICLR 2025)

---

### Meta-Review · Area_Chair_JdMX · 2025-12-29

**Summary:**

This paper addresses the "shallow alignment" problem in machine unlearning for LLMs, revealing that existing methods often suppress rather than erase target knowledge by introducing "spurious unlearning neurons." The authors propose a regularization-based method (SSIUU) to mitigate this issue. The submission received consistent praise for its novel problem formulation and mechanistic insights. During the rebuttal, the authors provided extensive clarifications and new experiments (including Integrated Gradients, MIA robustness, and computational cost analysis), which convincingly addressed the reviewers' main technical concerns.

**Reviewer Concerns:**

Addressed Concerns:

- Attribution Method Reliability: Reviewers (Hfxi, VL5g) questioned the reliance on a single attribution method. The authors successfully addressed this by providing additional experiments using Integrated Gradients, which yielded consistent findings, and by theoretically comparing their approach with other knowledge quantification methods.

- Computational Cost: Reviewers (Hfxi, A8GP, VL5g) were concerned about the overhead of calculating attributions at every step. The authors provided a detailed gradient derivation (using HVP) and a concrete runtime analysis table, demonstrating that the overhead is manageable (1.42x training time), effectively resolving this concern.

- Robustness to Privacy Attacks: Reviewer A8GP asked about robustness against other attacks like MIA. The authors clarified existing results and added new experiments on output logit-based MIA, showing their method's superiority over baselines.

- Conceptual Clarity: Questions regarding the definition of "neurons" and "spurious unlearning" were addressed with clear definitions and a commitment to include them in the final version.

Outstanding Concerns:

None. The authors have effectively addressed all substantive technical concerns raised by the reviewers.

**Reviewer Scores:**

- Reviewer VL5g (6 -> 6/8): This reviewer was already positive and stated that addressing the attribution and cost concerns would strengthen the paper. The authors' thorough rebuttal on these exact points likely warrants a score increase.

- Reviewer Hfxi (6 -> 6): Similar to VL5g, this reviewer's main concerns were methodological (attribution validity, implementation details). The comprehensive response, including gradient derivations and code explanations, would likely lead to a more confident acceptance.

- Reviewer A8GP (4 -> 6): The primary reason for the lower score was the lack of quantitative cost analysis. The authors provided a detailed table showing the training latency is acceptable, directly removing the main blocker for this reviewer.

- Reviewer 2ps9 (6 -> 6/8): This reviewer posed interesting questions about the causal role of neurons. The authors' new ablation experiment (removing top neurons) provided valuable empirical evidence supporting their hypothesis, further solidifying the paper's claims.

---

### Decision · Program_Chairs · 2026-01-26

Accept (Poster)